# GEOMETRICALLY ALIGNED TRANSFER ENCODER FOR INDUCTIVE TRANSFER IN REGRESSION TASKS

**Sung Moon Ko**[*], **Sumin Lee**[*], **Dae-Woong Jeong**[*], **Woohyung Lim**, **Sehui Han**
LG AI Research
{sungmoon.ko, sumin.lee, dw.jeong, w.lim, hansse.han}@lgresearch.ai

## ABSTRACT

Transfer learning is a crucial technique for handling a small amount of data that is potentially related to other abundant data. However, most of the existing methods are focused on classification tasks using images and language datasets. Therefore, in order to expand the transfer learning scheme to regression tasks, we propose a novel transfer technique based on differential geometry, namely the Geometrically Aligned Transfer Encoder (*GATE*). In this method, we interpret the latent vectors from the model to exist on a Riemannian curved manifold. We find a proper diffeomorphism between pairs of tasks to ensure that every arbitrary point maps to a locally flat coordinate in the overlapping region, allowing the transfer of knowledge from the source to the target data. This also serves as an effective regularizer for the model to behave in extrapolation regions. In this article, we demonstrate that *GATE* outperforms conventional methods and exhibits stable behavior in both the latent space and extrapolation regions for various molecular graph datasets.

## 1 INTRODUCTION

A machine learning model requires abundant data to learn and perform effectively. However, collecting sufficient amounts of data often consumes substantial amounts of energy and time. One of the most practical approaches to addressing this issue is to make use of large datasets correlated with the target data. Transfer learning stands out as one of the potential contenders. Transfer learning focuses on transferring knowledge from source to target data. Recently, there has been immense development in various domains such as languages (Zhuang et al., 2011; Long et al.; Zhuang et al., 2013; 2014; Pan et al., 2020), computer vision (Quattoni et al., 2008; Kulis et al., 2011; Raghu et al., 2019; Yu et al., 2022) and biomedical domain (Wang et al., 2019; Peng et al., 2021).

Despite these achievements, the primary area of concern is mostly limited to classification tasks in the vision and language domains. Moreover, identifying an architecture that effectively addresses regression problems has proven to be challenging. Hence, our primary objective is to develop an algorithm that can be generally applied to regression problems. In this article, our main focus centers on molecular property prediction tasks (Scarselli et al., 2009; Bruna et al., 2013; Duvenaud et al., 2015; Defferrard et al., 2016; Jin et al., 2018; Coley et al., 2019; Ko et al., 2023). Molecular datasets tend to be small in size, have a large number of task types, and are mainly regression. This makes molecular property prediction a good application to test our novel algorithm.

The fundamental concept of our method rests on a geometric interpretation of the encoding space. Given that the encoding process within a model involves intricate non-linear mappings, it is reasonable to regard the encoding space as non-linear and curved. As it is illustrated in Figure 1, we assume a data point in each task is on a manifold, and its collection formulates a coordinate patch that can be interpreted as a task coordinate. Each task coordinate has a chart to map to another correlated task to form a manifold. Additionally, machine learning models are obligated to exhibit continuity and differentiability to facilitate training through backpropagation. This requirement extends to the encoding space, which should ideally possess smoothness properties. These observations collectively suggest the possibility of interpreting the encoding space as a Riemannian manifold.

---

[*]These authors contributed equally to this work

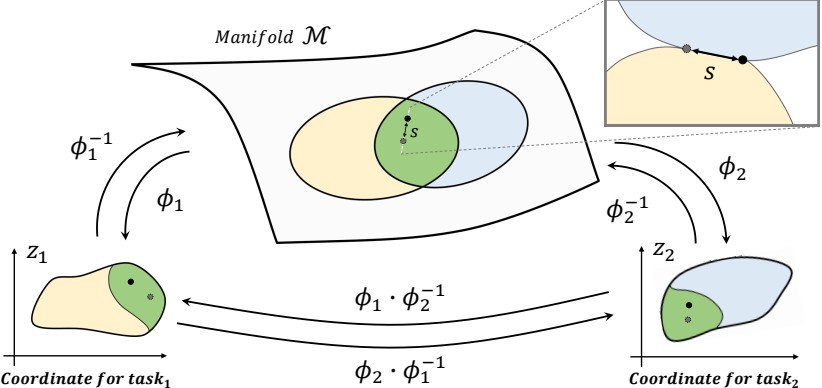

Figure 1: Two different coordinate frames are demonstrated in the figure, with coordinate transformation maps to each individual coordinate choice. One can interpret each coordinate frame as task-specific coordinates and map them with transformation models. An arbitrary point in the overlapping region of the manifold can be transformed from one task coordinate to another by combining mapping functions $\phi$. Moreover, by introducing perturbation points, as demonstrated in the figure, one can define the distance between points to match the geometrical shape in the overlapping region.

In general, a curved space is considerably more intricate than a flat space. For instance, when dealing with a curved space, determining the distance between points requires the computation of entire geodesic equations based on a specific metric. Consequently, the straightforward calculation of similarities between data points using Euclidean distance no longer applies, making it challenging to discern meaningful similarities between points. However, when the manifold is regarded as a Riemannian manifold, there are strategies that can help navigate these challenges. One such strategy involves finding diffeomorphisms that allow points to be situated in a locally flat frame (LF). As a result, it becomes feasible to develop models specifically designed to identify appropriate diffeomorphisms. These models aim to confine individual data points to LF within overlapping regions that align with other coordinate patches. This approach simplifies the process of establishing correspondence between similar points across different coordinate systems, as each point is effectively situated within a Euclidean environment on a local scale.

In essence, the act of matching overlapping points on a manifold ensures the alignment of their geometries. This alignment is crucial because the encoding space's inherent geometry in the source data contains relevant information for the target data. By achieving this alignment, the model facilitates the flow of information from the source to the target. In effect, this approach establishes a strong foundation for the model's capabilities and extends its capacity to generalize effectively.

Our main contribution of the article is as follows.

- We design a novel transfer algorithm (*GATE*) based on Riemannian differential geometry.
- The *GATE* is utilizable on regression tasks for inductive transfer learning.
- The *GATE* outperforms conventional methods in various molecular property regressions.
- The *GATE* exhibits stable underlying geometry and robust behavior in extrapolation tasks.

## 2 RELATED WORKS

### 2.1 TRANSFER LEARNING

Given a source domain $D^s$ with source task $T^s$ and a target domain $D^t$ with target task $T^t$, transfer learning aims to improve the learning of a better mapping function $f^t(\cdot)$ for the target task $T^t$ with the transferable knowledge gained from the $D^s$ and $T^s$ (Pan & Yang, 2009). Depending on the relationship between the source and target domains and tasks, transfer learning can be divided into three categories: inductive, transductive, and unsupervised transfer learning. Our problem setting is inductive transfer learning, where the target task is different from the source task ($T^s \neq T^t$),

regardless of whether the source and target domains are the same (Yang et al., 2020). Additionally, molecular property prediction is a regression task. In transductive transfer learning, studies have been done on regression tasks such as domain adaptation regression (DAR) (Chen et al., 2021). However, recent studies on regression tasks in an inductive transfer learning setting were only found in papers that applied it to a specific domain (Li & Fourches, 2020; Hoffmann et al., 2023).

## 2.2 MULTI-TASK LEARNING AND KNOWLEDGE DISTILLATION

Multi-task learning (MTL) can be viewed as a form of inductive transfer learning that learns multiple related tasks simultaneously. By using a shared representation, the knowledge gained from one task can help the learning of other tasks (Caruana, 1997). MTL and transfer learning both aim to generalize knowledge across different tasks. However, MTL aims to improve performance on a set of target tasks with equal importance, whereas transfer learning only focuses on one target task. Studies using MTL in various domains can be found (Lee & Kim, 2019; Liu et al., 2022).

Knowledge distillation (KD) is a training paradigm where the knowledge from a large teacher model is transferred to a small student model. According to (Gou et al., 2021), distillation techniques can be classified by knowledge types: response-based, feature-based, and related-based knowledge. Response-based knowledge, also called "dark knowledge", uses the output of the last layer, such as logits (Hinton et al., 2015). Feature-based knowledge distillation utilizes the output of intermediate layers, which are feature representations, to guide the student model. Relation-based KD transfers the relationships between features or data samples. For molecular property prediction, (Joshi et al., 2022) introduced feature-based distillation for GNNs by preserving global topology. Studies using cross-modal KD can also be found (Zhang et al., 2022; Zeng et al., 2023).

## 2.3 GEOMETRICAL DEEP LEARNING

Methods that extend deep neural networks to Euclidean and non-Euclidean domains, such as graphs and manifolds, are collectively referred to as geometric deep learning(GDL) (Bronstein et al., 2017). Regarding "how to transfer", transfer learning can be categorized into instance-based, feature-based, model-based, and relation-based transfer learning. A feature-based approach aims to find a feature representation that is effective for both the source and target domains. In the context of domain adaptation, the subspace spanned by the features in the source and target domains is the knowledge that is transferred (Yang et al., 2020). Manifold feature learning can learn tight representations that are invariant across domains. Some studies have used the property of manifold to perform unsupervised domain adaptation. By embedding datasets into Grassmann mainfolds, (Gopalan et al., 2011) obtained intermediate subspaces by sampling points along the geodesic between the source and target subspaces. (Gong et al., 2012) extended on this by learning a geodesic flow kernel (GFK) between domains. (Baktashmotlagh et al., 2014) embedded the probability distributions on a Riemannian manifold and used a Hellinger distance to approximate the geodesic distance. (Luo et al., 2020) proposed a Riemannain manifold embedding and alignment framework that used intra-class similarity and manifold metric alignment loss to achieve discriminability and transferability, respectively. However, to our best knowledge, we could not find geometrical deep learning approaches for inductive transfer learning in regression tasks.

## 3 GEOMETRICALLY ALIGNED TRANSFER ENCODER

The latent vector is believed to capture the essence of information for a given task. Therefore, one may consider if two different but correlated tasks may share a similar underlying geometry of latent spaces. Hence, if the latent spaces of given tasks can be aligned smoothly, mutual information will flow through the latent spaces from one another. This will lead to the superior performance of a target task not only in interpolation but also in extrapolation cases, which is the basic strategy of the *GATE*. However, deep learning models are designed to mimic non-linear functions by excavating information from the training dataset. The underlying geometry of models should inherently exhibit non-linearity and non-trivial curvature, as well as latent spaces, which are induced by non-linear models. This makes it challenging to match the underlying geometry of latent spaces.

There are several assumptions required to put geometrical concept into practice: 1) A pair of source and target tasks should be correlated; 2) a pair of tasks should have an overlapping region; 3) a

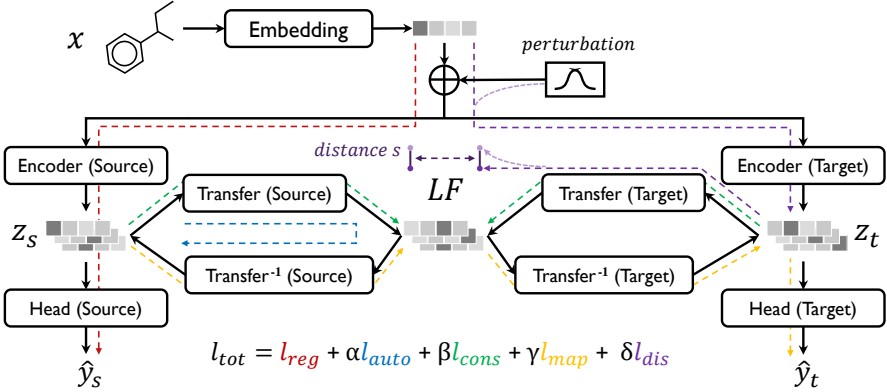

Figure 2: Schematic diagram for the *GATE* algorithm

mathematically well-defined underlying geometry is preferred, such as a Riemannian. Molecular datasets provide a perfect proving ground since they satisfy the first two assumptions, and with some proper constraints, the third assumption can also be established. First, there exist numerous sets of correlated tasks proven by scientific research. Also, the majority of molecules possess values for multiple properties, which ensures a large area of overlapping regions across a pair of tasks. Finally, by imposing the mathematical characters of Riemannian geometry as constraints, the underlying geometry can be interpreted as a Riemannian manifold. Moreover, every molecule can be expressed in a single corresponding universal string in, for instance, the SMILES format(Weininger, 1988).

## 3.1 RIEMANNIAN GEOMETRY

In Figure 2 we first take an input SMILES and embed it into the corresponding vector. After embedding, latent space is formulated by encoders, which consist of DMPNN(Yang et al., 2019) and MLP layers. The latent vector is fed into task-corresponding heads for inference properties. Here we utilize MSE for basic regression loss in the training scheme as follows:

$$l_{\text{reg}} = \frac{1}{N} \sum_i^N (y_i - \hat{y}_i)^2 \tag{1}$$

Where $N$, $y_i$, and $\hat{y}_i$ are the number of data points, target, and predicted value, respectively. This is a simple, basic loss to train a molecular property, yet to match the latent space and design a transfer scheme requires more mathematically advanced techniques.

The Riemannian manifold enjoys isometries induced by diffeomorphisms, which guarantee freedom of coordinate choices on the manifold. Thus, one can find a LF around a point on a manifold. A LF means the coordinate patch can be interpreted as Euclidean around its infinitesimal boundary. Note that the input vector can be embedded on a smooth manifold, and its latent space can also be assumed to be smooth since the ML model should always be smooth at an arbitrary point due to the backpropagation training scheme. Hence, by utilizing the model to map encoder space to a LF, the inner product of vectors satisfies a positive semi-definite property. Considering these aspects, we can fix the latent space to a Riemannian. This whole process is closely described in later sections.

## 3.2 MATCHING COORDINATE PATCHES

As we mentioned earlier, we interpret the latent vector from the encoder model to be on a Riemannian manifold. Moreover, we assume each downstream task corresponds to a specific coordinate patch on the manifold with a vast overlapping region in molecular property prediction cases. Since one can always find a mapping relation between different coordinate patches, it is crucial to find the specific analytic form of the mapping function. Yet, it is almost impossible to find an analytic form of the mapping function of an unknown manifold with little numerical information. Hence, we utilize a deep learning model to learn the mapping relation instead of finding one with hands.

The coordinate mapping is induced by a Jacobian on a point. Since a vector enjoys isometry on diffeomorphism on a Riemannian manifold and is written in fundamental representation, the coordinate transformation can be expressed as follows:

$$z'^i \equiv \sum_j \frac{\partial z'^i}{\partial z^j} z^j \tag{2}$$

By introducing a model that mimics the above Jacobian and predicts the transformation matrix for a given vector, one can freely transform between coordinates. However, its inverse may not be stable, and moreover, its derivative may also not be stable and cause instability in the backpropagation process. Resolving the issue is critical for training the model. Therefore, we tweak the problem to not just mimic the Jacobian itself but to predict a transformed vector. Furthermore, we utilize an autoencoder scheme to ensure its inverse behavior is accurate and stable.

$$z' = \text{Transfer}(z), \qquad \hat{z} = \text{Transfer}^{-1}(z') \tag{3}$$

We indeed utilize MSE loss for the autoencoder which consists of transfer and its inverse modules.

$$l_{\text{auto}} = \text{MSE}(z, \hat{z}) \tag{4}$$

Here, we restrict the model by imposing an autoencoder loss to match the overlapping region of coordinates. But this is insufficient; thus, stronger conditions are necessary to match coordinates.

Since we are expecting transformations to occur between coordinate patches, the transfer and inverse modules should have the same input and output dimensions. Moreover, as we are trying to match the underlying geometry of overlapping regions, which are in fact the latent space of the autoencoder, one can realize that we can require another condition for the autoencoder. We consider two distinct autoencoders for source and target coordinates that map coordinate patches to same LF back and forth; the transfer output from both should be equal in order to match the geometry of the overlapping region.

$$z'_s = \text{Transfer}_{s \to LF}(z_s) \qquad \hat{z}_s = \text{Transfer}^{-1}_{LF \to s}(z'_s) \tag{5}$$

$$z'_t = \text{Transfer}_{t \to LF}(z_t) \qquad \hat{z}_t = \text{Transfer}^{-1}_{LF \to t}(z'_t) \tag{6}$$

Where $\text{Model}_{s \to LF}$ indicate a model from source to LF frame and vice versa.

$$l_{cons} = \text{MSE}(z'_s, z'_t) \tag{7}$$

This loss, we call it a consistency loss, pivots points on overlapping regions to glue the underlying geometry. Furthermore, one can induce information flow from the source domain to the target domain by utilizing a mixture of autodencoders.

$$z'_s = \text{Transfer}_{s \to LF}(z_s) \qquad \hat{z}_{s \to t} = \text{Transfer}^{-1}_{LF \to t}(z'_s) \tag{8}$$

The above equation indicates a vector from the source domain that is transformed to the target domain, retaining its mutual information. Therefore, this mapped vector $\hat{z}_{s \to t}$ should also predict the same label as the original vector $z_t$ on the target side. This fact gives rise to another form of loss function, namely a mapping loss.

$$l_{map} = \text{MSE}(y_t, \hat{y}_{s \to t}) \tag{9}$$

Here $y_t$ indicates the label for corresponding vectors and $\hat{y}_{s \to t}$ indicates predicted value from $\hat{z}_{s \to t}$. Above loss ensures mutual information flow by matching overlapping coordinates on pivot points. This particular loss is applied to reverse case too which maps $z_t$ to $\hat{y}_{t \to s}$.

Now the question is, 'Is this enough?'. The answer is 'No!'. The reason is actually not intricate to find. If we start with immensely much data on both the source and target domains, then the answer will be 'Yes'. However, in most cases, we lack data points to glue coordinates together. If points are not enough, there are always overfitting issues that cause the geometry to crumple outside the pivot point region. There are two solutions to this issue. One is gathering more data points and hoping to work smoothly, and the other is by adding a stronger condition to the geometry.

Our idea is not only to focus on point data but also on its geometries by considering the geodesics of two different points. Here, there are two issues to compute the geodesics in curved space: one is that the space itself is curved, and the Pythagorean theorem no longer holds on the space in general.

The other is that the shortest path between two points in a curved space is no longer intuitive. In curved space, one can calculate distance using the following equation:

$$S^2 = \int_l \sum_\mu \sum_\nu g_{\mu\nu} dx^\mu dx^\nu \tag{10}$$

Here, $g_{\mu\nu}$ is a metric on the surface, $dx^\mu, dx^\nu$ are infinitesimal displacements, and $l$ is the geodesic path between two points. The geodesic of a curved space can be defined as a curve whose parallel transport along the curve preserves the tangent vector to the curve. Which indicates,

$$\nabla_{\dot\gamma(t)} \dot\gamma(t) = 0 \tag{11}$$

Where $\gamma(t)$ is a curve on a manifold, $\dot\gamma(t)$ is the derivative respect to a parameter $t$, and $\nabla$ is a covariant derivative with an affine connection $\Gamma^\mu_{\nu\rho}$. It is clear that solving the geodesic equation is not a simple task since one must know the explicit form of the metric of space, and the equation itself is a non-linear partial differential equation. Thus, as depicted in Figure 2, we introduce infinitesimal perturbation points around a data point to detour the issue. The infinitesimal perturbation makes things easy to tackle since a data point can always be expressed on a LF, where a geodesic becomes trivial. Then one can easily compute the distance between a data point and perturbed points since it can be interpreted to be on a flat Euclidean space.

The distance between two different points on a space is a key quantity to match the task coordinates since, as shown in eq. 10, distance always requires metric on a space. By restricting neighbor points to an infinitesimal region around a pivot data point, the metric can now be considered a Euclidean metric, $\eta_{\mu\nu}$. Than the eq. 10 simplified as follows:

$$S^2 = \int_l \sum_\mu \sum_\nu g_{\mu\nu} dx^\mu dx^\nu = \int_l \sum_\mu \sum_\nu \eta_{\mu\nu} dx^\mu dx^\nu = \int_a^b dx^2 \tag{12}$$

Where $a$ and $b$ are the pivot data point and its perturbation point, respectively. If perturbation is infinitesimal, one can take off the integral and find displacement in a simplified form.

$$S = |b - a| \tag{13}$$

Since it is possible to find a LF on a Riemannian manifold, for a given data point, two different task coordinates can always be transformed by their diffeomorphisms to be on a flat coordinate. If we utilize the above distance between pivot and perturbation points, each task coordinate is now glued together by not only a given pivot point but also by their neighbors. It is closely depicted in Figure 2. Right after the embedding, we find perturbation points around given data points and feed them to each task-specific model for encoding. After encoding, models will be entangled together by mapping loss, consistency loss, and distance loss, forcing the distances between pivot points and their perturbations to be equal between tasks.

$$l_{dis} = \frac{1}{M} \sum_i^M \text{MSE}(s_s^i, s_t^i) \tag{14}$$

Where $M$ is the number of perturbations and $s_s^i$ is the displacement between pivot data points and their perturbations.

$$s_s^i \equiv |(z_s') - (z_s'^i)| \qquad s_t^i \equiv |(z_t') - (z_t'^i)| \tag{15}$$

$$z_s'^i = \text{Transfer}_{s \to LF}(\text{Encoder}_s(x^i)) \tag{16}$$

$$z_t'^i = \text{Transfer}_{t \to LF}(\text{Encoder}_t(x^i)) \tag{17}$$

Here $x^i$ denotes $i$th perturbation of embedded $x$, and $\text{Encoder}_s$ and $\text{Encoder}_t$ are encoder parts of source and target model respectively. Finally, by gathering all losses with individual hyperparameters, we obtain the complete form of the loss function used in the *GATE* algorithm.

$$l_{tot} = l_{reg} + \alpha l_{auto} + \beta l_{cons} + \gamma l_{map} + \delta l_{dis} \tag{18}$$

Each hyperparameter tunes the corresponding weight of loss. In most cases, it is sufficient to let all the hyperparameters to 1, but in some cases, one can fine-tune the hyperparameters to achieve superior performance. The most critical hyperparameter is $\delta$ since the distance loss manages the regularization effect of the model. This result will be closely examined in the ablation section. And with the proper hyperparameters, our novel transfer model *GATE* outperforms conventional models in inductive transfer settings for regression tasks, as shown in the experiment section.

## 4 EXPERIMENTS

### 4.1 EXPERIMENTAL SETUP

To test our algorithms, we used a total of 14 datasets from three different open databases named PubChem(Kim et al., 2022), Ochem(Sushko et al., 2011), and CCCB(III, 2022), as described in Appendix Table 3. The training and test split is an 80:20 ratio, and two sets of data were prepared: random split and scaffold-based split(Bemis & Murcko, 1996). Every experiment is tested in a four-fold cross-validation setting with uniform sampling for accurate evaluation, and a single NVIDIA A40 is used for the experiments. For the evaluation, we compare the performance of *GATE* against that of single task learning (STL), MTL, KD, global structure preserving loss based KD (GSP-KD) (Joshi et al., 2022), and transfer learning (retrain all or head network only). We used the same architecture for encoders and heads in both the baselines and our model for all experiments. More detailed experimental setups can be found in Appendix. We performed experiments with a total of 23 target and source task pairs, respectively.

### 4.2 RESULTS

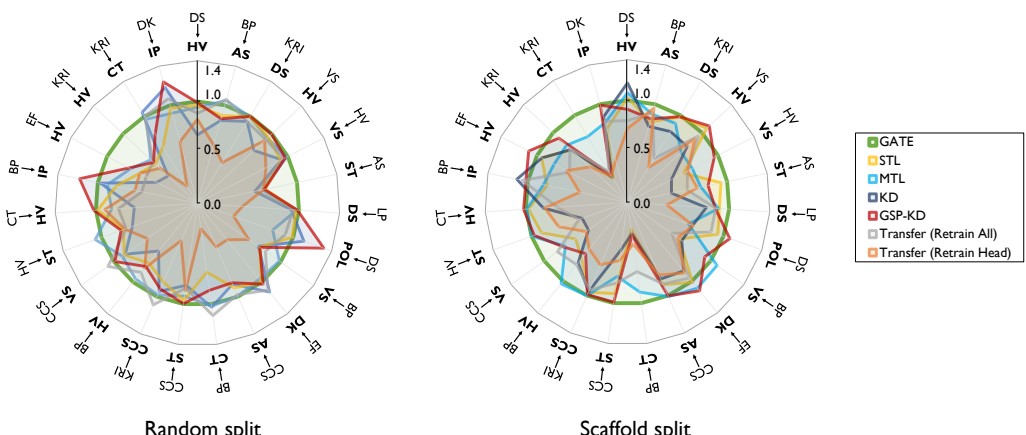

Figure 3: The RMSE of *GATE* divided by the RMSE of the corresponding methods are illustrated in the radar chart. Experiments were performed with 23 target and source task pairs in random split (left) and scaffold split (right) datasets. Inner characters correspond to the target tasks, and outer characters are for the source tasks. The full names of the tasks can be found in Appendix Table 3.

The result of the training for the 23 task pairs with both random and scaffold split is demonstrated in Figure 3. The figure illustrates the relative regression accuracy, calculated as the RMSE of *GATE* divided by the RMSE of the corresponding method. As shown in the figure, in the majority of cases, *GATE* outperforms baseline methods by significant margins. Note that, even though the KD and GSP-KD outperforms the *GATE* for some task pairs, they show critically low accuracy for several task pairs. On the other hand, as we can see in the Appendix Table 4-7, *GATE* exhibited the best performance in more than half of task pairs, specifically 12 pairs in random split and 13 pairs in scaffold split out of 23 pairs. Considering that the method with the second-highest number of best-performing task pairs had only 4 task pairs in random split and 5 task pairs in scaffold split, this can be regarded as a dramatic difference. Also, when considering the second-best performance, *GATE* maintained consistently high performance in the majority of pairs, reaching 18 pairs in random split and scaffold split. For the average RMSE over all task pairs, *GATE* demonstrates significantly lower RMSE compared to the second-best method, 9.8 % lower in random split (GSP-KD) and 14.3 % lower in scaffold split (MTL). This indicates that *GATE* performs even better in extrapolation tasks.

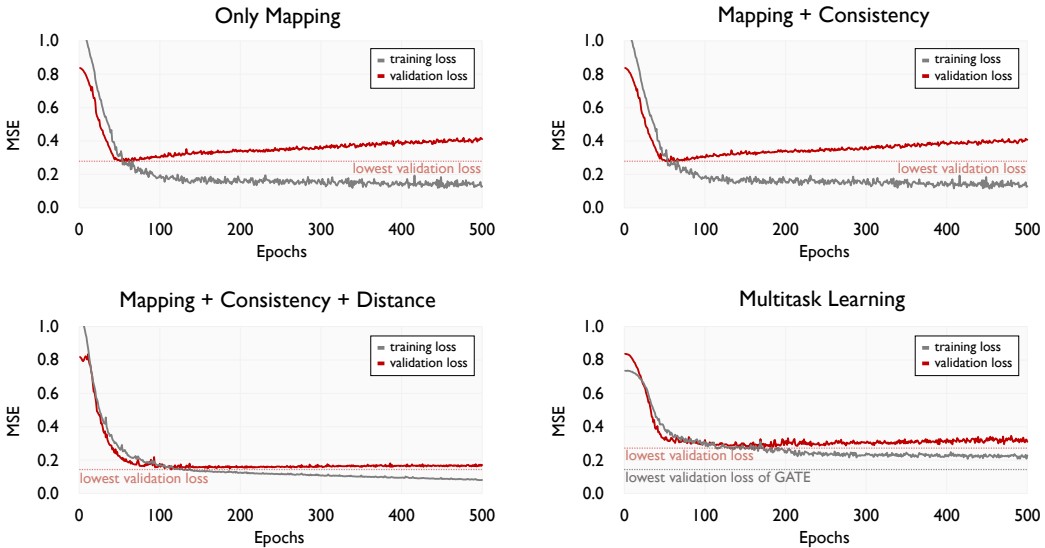

Figure 4: Training and validation loss curves of *GATE* and MTL models with different sets of loss terms. Upper left: *GATE* trained without consistency and distance loss; upper right: *GATE* trained without distance loss; lower left: plain *GATE*, lower right: plain MTL.

## 5 ABLATIONS STUDIES AND FURTHER ANALYSIS

### 5.1 ROLE OF DISTANCE LOSS

The most distinguishable aspect of *GATE* is the distance loss, which connects task coordinates not only through labeled data points but also surrounding perturbation points. Since learning signals from consistency and distance loss do not require regression target labels, these two losses can be applied in an unsupervised manner. *GATE* directly leverages this advantage in distance loss and regularizes the overall latent space to be geometrically transferable to a LF. To verify this, we trained *GATE* using different sets of losses: 1) only mapping, 2) mapping and consistency, and 3) mapping, consistency, and distance losses. As a result, we observed that the addition of distance loss significantly suppresses overfitting during the training process, as shown in Figure 4. Note that the training loss curve of MTL never reached the lowest validation loss of *GATE* (grey dashed line). These results indicate that the regression model can be strongly regularized by the distance loss while having a minimal adverse impact on the regression task itself.

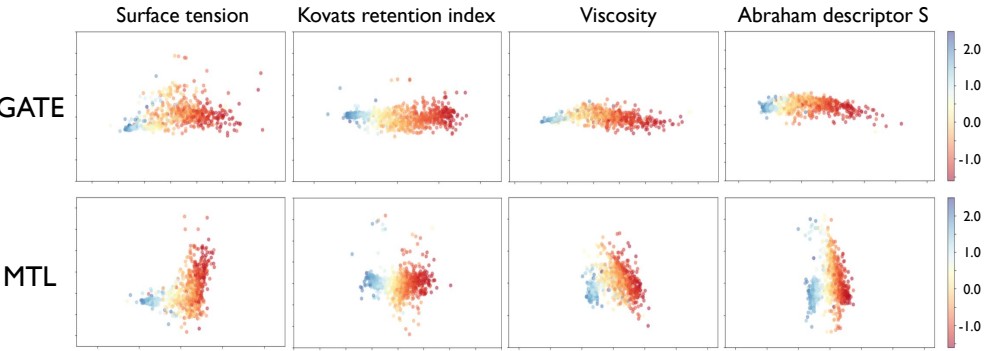

Figure 5: PCA on latent space of *GATE* and MTL model trained with collision cross section dataset as a target task and surface tension, Kovats retention index, viscosity, and Abraham descriptor S as source tasks, respectively. The color indicates the predicted value of the collision cross section.

## 5.2 STABILITY OF LATENT SPACE

For further investigation on the regularization effect of *GATE*, we examined the shape of the latent space of the *GATE* and MTL models. Transfer learning offers a way to leverage information from source tasks, but undesired interfering information can cause negative transfer. Ideally, if a model is well-guided by the right information and regularized properly, the overall geometry of the latent space may remain stable and not depend on the type of source tasks. (Zhang et al., 2023; Gómez-Bombarelli et al., 2018) However, if the target task is overwhelmed by the source task and regularization is not enough, latent space will be deformed according to the source tasks. To check this effect, we trained regression on the same target task using *GATE* or MTL with four different source tasks and then performed principal component analysis. Figure 5 effectively illustrates that the latent space of *GATE* exhibits relatively stable characteristics compared to that of MTL.

## 5.3 ROBUSTNESS TO DATA CORRUPTION

We examined the robustness against data corruption in order to directly verify the regularization effect. To artificially introduce significant corruption to the dataset, data points with labeled values outside the standard deviation were randomly selected, and the values of these data points were altered to be twice the standard deviation with negation. And then, after training with the corrupted dataset, we assessed the mean squared error (MSE) between the model's predictions on the corrupted data points used during training and the original values before the corruption. If the model exhibits overfitting to the corrupted data, it will manifest substantial errors, whereas effective model regularization will lead to error reduction. Figure 6 shows the result of the test for *GATE* and MTL. The *GATE* model exhibits considerably lower errors compared to MTL. This result indicates that the regularization effect of the *GATE* algorithm leads to high robustness against data corruption and successful generalization.

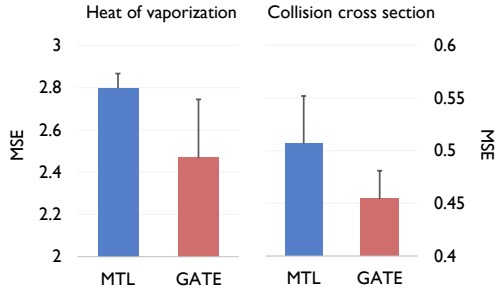

Figure 6: *GATE* and MTL models are trained with artificially corrupted data points and MSEs between predicted values and original values before the corruption is recorded. The target task is heat of vaporization (left) or collision cross-section (right), with the opposite side being used as the source task.

## 6 DISCUSSION

We demonstrated a novel transfer algorithm based on the geometrical interpretation of latent space. By assuming latent space to be on a Riemannian manifold with a curved metric, the conventional Euclidean interpretation is no longer valid. However, since diffeomorphism invariance guarantees freedom of coordinate choices, one can always find a LF for each task coordinate. Hence, it is possible to impose consistency loss by matching an encoded vector from one task to another. Furthermore, we showed that the distance loss of displacement from a point in an overlapping region to its perturbations is crucial. All together, we demonstrated that the *GATE* outperforms conventional methods in the majority of molecular prediction task setups and proved its robustness.

In this article, we focused on regression tasks for molecular property prediction. Yet, the algorithm itself is not restricted to regression tasks. Thus, it is interesting to expand the algorithm to other domains such as images, languages, or, further, multi-modal setups. Another technical point is that we considered the neighboring region of a pivoting point to compute displacement on a flat space. However, to match the entire geometry, it is necessary to consider not only perturbations but also points in finite-distance regions. In this scheme, one must solve a geodesic equation instead of merely computing the Euclidean distance. It is incomparably complicated since one must consider 1) finding an explicit form of curved metric and 2) solving highly non-linear differential geodesic equations analytically or numerically. Yet, considering *GATE*'s performance, by imposing geometric characters on fully curved backgrounds, the performance gains may be substantial. Hence, this could be another interesting research topic to expand *GATE*'s horizons.

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

# A   NOTATIONS

Our notation is based on index notation and Einstein summation conventions. Notation of functions and matrices in our algorithm is as follows.

$$X : \text{Vector}$$
$$X^\mu : \text{Vector Field}$$
$$dx_\mu : \text{Basis}$$
$$X_\mu : \text{Dual Vector Field}$$
$$dx^\mu : \text{Dual Basis}$$
$$T : \text{Tensor}$$
$$T^{\nu_1 \cdots \nu_p}{}_{\mu_1 \cdots \mu_q} : \text{(p, q) Tensor Field}$$
$$g_{\mu\nu} : \text{Metric Tensor}$$
$$\delta_{\mu\nu} : \text{Kronecker Delta}$$
$$\nabla_\mu : \text{Covariant Derivative}$$
$$\mathcal{L}_X : \text{Lie Derivative}$$
$$\Gamma^\rho{}_{\mu\nu} : \text{Christoffel Symbol}$$

All indices are raised and lowered by a metric $g_{\mu\nu}$. For instances,

$$g^\mu{}_\nu = g^{\mu\rho} g_{\rho\nu} \tag{19}$$

where

$$g^{\mu\nu} g_{\mu\nu} = \delta^\mu{}_\nu = D \tag{20}$$

Here $D$ is the number of dimensions.

# B   PROOFS AND DERIVATIONS

## B.1   THE DEFINITION OF RIEMANNIAN MANIFOLD

A curved space is complicated to comprehend in general. Since late 19th century, there has been immense development in differential geometry to interpret curved spaces formally. One of the best-known intuitive geometrys is the Riemannian. Riemannian geometry enjoys a handful of useful mathematical characters that can be utilized in the real world. The formal definition of Riemannian is as follows:

**Definition B.1** (Riemannian Manifold). A Riemannian metric on a smooth manifold M is a choice at each point $x \in M$ of a positive definite inner product $g_p : T_pM \times T_pM \to \mathbb{R}$ on $T_xM$. The smooth manifold endowed with the metric $g$ is a Riemannian manifold, denoted $(M, g)$.

As it is expressed above, a Riemannian manifold is smooth and differentiable everywhere on the manifold and its derivative as well. Also, a Riemannian enjoys diffeomorphism invariances, induced by the Lie derivative $\mathcal{L}_X$. One can easily notice that the adjoint operation between two different Lie derivatives forms a group, namely the diffeomorphism group. This isometry ensures coordinate choices without changing the global geometry of the space.

$$X' = X'^\mu dX'_\mu = X'^\mu \frac{\partial X^\nu}{\partial X'^\mu} dX_\nu = X^\nu dX_\nu = X \tag{21}$$

As it is depicted in eq. 21, transformed vector remains unchanged. Moreover, one can always fix the transformed coordinate in a locally flat space.

$$\xi^\mu = \frac{\partial \xi^\mu}{\partial X^\nu} X^\nu \tag{22}$$

Where $\xi^\mu$ is a vector on a locally flat frame. To ensure the vector is on a flat frame, one must impose the following condition:

$$\frac{\partial^2}{\partial t^2} \xi^\mu(t) \equiv 0 \tag{23}$$

Since a vector is on a flat frame, it should be in a free-falling motion, so its acceleration should be trivial. On a locally flat frame, the metric also becomes flat Euclidean metric

$$g_{\mu\nu} = 1_{\mu\nu} \tag{24}$$

### B.2 COVARIANCE

The vector should be transformed in the same manner in any coordinate frame. However, if the space is no longer flat, the ordinary derivative no longer guarantees it. Let us consider a derivative of a vector in a general curved space.

$$\partial_\mu \to \partial'_\mu = \frac{\partial x^\mu}{\partial x'^\nu} \partial_\nu \tag{25}$$

Where $\partial_\mu = \frac{\partial}{\partial x^\mu}$, then the vector transformation can be written as follows:

$$\partial_\nu X^\mu \to \partial'_\nu X'^\mu = \frac{\partial x^\lambda}{\partial x'^\nu} \frac{\partial}{\partial x^\lambda} \left( \frac{\partial x'^\mu}{\partial x^\rho} V^\rho \right) \tag{26}$$

$$= \frac{\partial x'^\nu}{\partial x^\lambda} \left( \frac{\partial x'^\rho}{\partial x^\nu} \partial^\lambda V^\rho + \frac{\partial^2 x'^\mu}{\partial x^\lambda \partial x^\rho} V^\rho \right) \tag{27}$$

As it is shown above, a transformation of a vector on a curved space with an ordinary derivative is no longer covariant. Thus, one must impose an additional factor to make it covariant, namely an Affine connection. With this factor, one can define a covariant derivative, replacing an ordinary one.

$$\nabla_\mu = \partial_\mu + \Gamma^\lambda{}_{\mu\nu} \tag{28}$$

By requiring a covariance condition on the covariant derivative,

$$\nabla_\lambda \to \nabla'_\lambda V'^\mu = \frac{\partial x^\rho}{\partial x'^\nu} \frac{\partial x'^\mu}{\partial x^\nu} \nabla_\rho V^\nu \tag{29}$$

Then one can induce the explicit form of a connection.

$$\nabla_\mu V^\nu = \partial_\mu V^\nu + \Gamma^\nu{}_{\mu\lambda} V^\lambda \tag{30}$$

Under coordinate transformation,

$$\frac{\partial}{\partial x'^\mu} \left( \frac{\partial x'^\nu}{\partial x^\lambda} V^\lambda \right) + \Gamma'^\nu{}_{\mu\sigma} V'^\sigma = \frac{\partial x^\rho}{\partial x'^\mu} \frac{\partial x'^\nu}{\partial x^\lambda} \partial_\rho V^\lambda + \frac{\partial x^\rho}{\partial x'^\mu} \frac{\partial^2 x'^\nu}{\partial x^\rho \partial x^\lambda} V^\lambda + \Gamma'^\nu{}_{\mu\sigma} V'^\sigma \tag{31}$$

Here, to make the derivative of a vector covariant, the following equation must hold:

$$\frac{\partial x^\rho}{\partial x'^\mu} \frac{\partial^2 x'^\nu}{\partial x^\rho \partial x^\lambda} V^\lambda + \Gamma'^\nu{}_{\mu\sigma} V'^\sigma = \frac{\partial x^\rho}{\partial x'^\mu} \frac{\partial x'^\nu}{\partial x^\lambda} \Gamma^\lambda{}_{\rho\sigma} V^\sigma \tag{32}$$

Which is

$$\Gamma'^\nu{}_{\mu\sigma} \left( \frac{\partial x'^\sigma}{\partial x^\tau} V^\tau \right) = \frac{\partial x^\rho}{\partial x'^\mu} \frac{\partial'^\nu}{\partial x^\lambda} \Gamma^\lambda{}_{\rho\sigma} V^\sigma - \frac{\partial x^\rho}{\partial x'^\mu} \frac{\partial x^\rho}{\partial x'^\mu} \frac{\partial^2 x'^\nu}{\partial x^\rho \partial x^\lambda} V^\lambda \tag{33}$$

$$\Gamma'^\nu{}_{\mu\kappa} V^\tau = \frac{\partial x^\rho}{\partial x'^\kappa} \frac{\partial x^\rho}{\partial x'^\mu} \frac{\partial x'^\nu}{\partial x^\lambda} \Gamma^\lambda{}_{\rho\sigma} V^\sigma - \frac{\partial x^\tau}{\partial x'^\kappa} \frac{\partial x^\rho}{\partial x'^\mu} \frac{\partial^2 x'^\nu}{\partial x^\rho \partial x^\lambda} V^\lambda \tag{34}$$

This leads us to the explicit form of how the Christoffel symbol transforms under coordinate changes.

$$\Gamma'^\nu{}_{\mu\kappa} = \frac{\partial x^\tau}{\partial x'^\kappa} \frac{\partial x^\rho}{\partial x'^\mu} \frac{\partial x'^\nu}{\partial x^\lambda} \Gamma^\lambda{}_{\rho\tau} - \frac{\partial x^\tau}{\partial x'^\kappa} \frac{\partial x^\rho}{\partial x'^\mu} \frac{\partial^2 x'^\nu}{\partial x^\rho \partial x^\tau} \tag{35}$$

Since the Kronecker delta is a constant matrix, it is obvious that the derivative of the delta should be trivial. Then one can apply the chain rule to the delta and find the following relation, which can simplify the above transformation rule.

$$\frac{\partial}{\partial x'^\mu} \delta^\nu_\kappa = \frac{\partial}{\partial x'^\mu} \frac{\partial x'^\nu}{\partial x'^\kappa} = \frac{\partial}{\partial x'^\mu} \left( \frac{\partial x^\tau}{\partial x'^\kappa} \frac{\partial x'^\nu}{\partial x^\tau} \right) = 0 = \frac{\partial x^\tau}{\partial x'^\kappa} \frac{\partial x^\rho}{\partial x'^\mu} \frac{\partial^2 x'^\nu}{\partial x^\rho \partial x^\tau} + \frac{\partial x'^\nu}{\partial x^\tau} \frac{\partial x'^\nu}{\partial x^\tau} \frac{\partial^2 x^\tau}{\partial x'^\mu \partial x'^\rho} \tag{36}$$

Finally, the transformation rule for a Christoffel symbol is as follows:

$$\Gamma'^{\nu}_{\ \mu\kappa} = \frac{\partial x^{\tau}}{\partial x'^{\kappa}} \frac{\partial x^{\rho}}{\partial x'^{\mu}} \frac{\partial x'^{\nu}}{\partial x^{\lambda}} \Gamma^{\lambda}_{\ \rho\tau} + \frac{\partial x'^{\nu}}{\partial x^{\tau}} \frac{\partial^2 x^{\tau}}{\partial x'^{\mu} \partial x'^{\rho}} \tag{37}$$

By the same logic, one can easily find out how covariant derivatives act on forms.

$$\nabla_{\mu} V_{\nu} = \partial_{\mu} V_{\nu} - \Gamma^{\lambda}_{\ \mu\nu} V_{\lambda} \tag{38}$$

### B.3 EXPLICIT FORM OF CHRISTOFFEL SYMBOL

The metric is a ruler of a given geometry; it should not vary under position on a coordinate. The Euclidean is trivial to see since the metric on Euclidean space is mere $\delta_{\mu\nu}$, which is a constant matrix.

$$\frac{\partial}{\partial x^{\lambda}} \delta_{\mu\nu} = 0 \tag{39}$$

However, in the curved case, the above statement should also hold to interpret the metric as a ruler, yet the statement does not hold for an ordinary derivative. There, the covariant derivative kicks in to replace an ordinary derivative instead. By taking covariant derivative to the curved metric, the term diminishes.

$$\nabla_{\lambda} g_{\mu\nu} = 0 \tag{40}$$

One can express this in terms of a flat metric with a diffeomorphism transformation factor.

$$g_{\mu\nu}(x) = \frac{\partial \xi^{\lambda}}{\partial x^{\mu}} \frac{\partial \xi^{\rho}}{\partial x^{\nu}} \delta_{\lambda\rho}(\xi) \tag{41}$$

If we take a derivative of x on both sides, the above equation becomes:

$$\frac{\partial}{\partial x^{\sigma}} g_{\mu\nu}(x) = \frac{\partial^2 x^{\lambda}}{\partial x^{\sigma} \partial x^{\mu}} \frac{\xi^{\rho}}{\partial x^{\nu}} \delta_{\lambda\rho} + \frac{\partial^2 \xi^{\rho}}{\partial x^{\sigma} \partial x^{\nu}} \frac{\partial \xi^{\lambda}}{\partial x^{\mu}} \delta_{\lambda\rho} \tag{42}$$

$$= \frac{\partial^2 \xi^{\rho}}{\partial x^{\sigma} \partial x^{\nu}} \frac{\partial x^{\tau}}{\partial \xi^{\rho}} \frac{\partial \xi^{\rho}}{\partial x^{\tau}} \frac{\partial \xi^{\lambda}}{\partial x^{\mu}} \delta_{\lambda\rho} + \frac{\partial^2 \xi^{\lambda}}{\partial x^{\sigma} \partial x^{\mu}} \frac{\partial x^{\tau}}{\partial \xi^{\lambda}} \frac{\partial \xi^{\lambda}}{\partial x^{\tau}} \frac{\partial \xi^{\rho}}{\partial x^{\nu}} \delta_{\lambda\rho} \tag{43}$$

$$= \frac{\partial^2 \xi^{\rho}}{\partial x^{\sigma} \partial x^{\nu}} \frac{\partial x^{\tau}}{\partial \xi^{\rho}} g_{\mu\tau} + \frac{\partial^2 \xi^{\lambda}}{\partial x^{\sigma} \partial x^{\mu}} \frac{\partial x^{\tau}}{\partial \xi^{\lambda}} g_{\tau\nu} \tag{44}$$

From eq 40, one can easily find out the specific form of the Christoffel symbol in terms of derivatives of curved and flat coordinates.

$$\frac{\partial}{\partial x^{\sigma}} g_{\mu\nu} = \Gamma^{\tau}_{\ \sigma\mu} g_{\tau\nu} + \Gamma^{\tau}_{\ \nu\sigma} g_{\mu\sigma} \tag{45}$$

$$\Gamma^{\tau}_{\ \sigma\mu} = \frac{\partial^2 \xi^{\lambda}}{\partial x^{\sigma} \partial x^{\mu}} \frac{\partial x^{\tau}}{\partial \xi^{\lambda}}(x) \tag{46}$$

Since the metric should always be symmetric, the lower indices of the Christoffel symbol should also be symmetric. It is called a torsion-free condition. Furthermore, by utilizing a simple mathematical trick, one can obtain the Christoffel symbol in terms of the metric $g_{\mu\nu}$.

$$\frac{\partial}{\partial x^{\sigma}} g_{\mu\nu} = \Gamma^{\tau}_{\ \sigma\mu} g_{\tau\nu} + \Gamma^{\tau}_{\ \sigma\nu} g_{\mu\tau} \tag{47}$$

$$\frac{\partial}{\partial x^{\mu}} g_{\nu\sigma} = \Gamma^{\tau}_{\ \mu\nu} g_{\tau\sigma} + \Gamma^{\tau}_{\ \mu\sigma} g_{\nu\tau} \tag{48}$$

$$\frac{\partial}{\partial x^{\nu}} g_{\sigma\mu} = \Gamma^{\tau}_{\ \nu\sigma} g_{\tau\mu} + \Gamma^{\tau}_{\ \nu\mu} g_{\sigma\tau} \tag{49}$$

Adding the first two equations and subtracting the last one leads to

$$\Gamma^{\lambda}_{\ \mu\nu} = \frac{1}{2} g^{\lambda\rho} \left( \frac{\partial}{\partial x^{\mu}} g_{\nu\rho} + \frac{\partial}{\partial x^{\nu}} g_{\rho\mu} - \frac{\partial}{\partial x^{\rho}} g_{\mu\nu} \right) \tag{50}$$

### B.4 Geodesic Equations

The shortest path between two points is simple in flat space. However, in curved space, the notion becomes rather complicated. The shortest path in a curved space is defined as a geodesic. There are several ways to induce a geodesic equation. One is by requiring a free-falling condition.

$$\frac{\partial^2 \xi^\mu(\tau)}{\partial \tau^2} = 0 \tag{51}$$

By diffeomorphism, one can transform a coordinate into an arbitrary coordinate x.

$$0 = \frac{\partial}{\partial \tau}\left(\frac{\partial \xi^\mu}{\partial x^\nu}\frac{\partial x^\nu}{\partial \tau}\right) = \frac{\partial \xi^\mu}{\partial x^\nu}\frac{\partial^2 x^\nu}{\partial \tau^2} + \frac{\partial^2 \xi^\mu}{\partial x^\lambda \partial x^\nu}\frac{\partial x^\lambda}{\partial \tau}\frac{\partial x^\nu}{\partial \tau} \tag{52}$$

$$\frac{\partial^2 x^\rho}{\partial \tau^2} + \frac{\partial^2 \xi^\mu}{\partial x^\lambda \partial x^\nu}\frac{\partial x^\rho}{\partial \xi^\mu}\frac{\partial x^\lambda}{\partial \tau}\frac{\partial x^\nu}{\partial \tau} = \frac{\partial^2 x^\rho}{\partial \tau^2} + \Gamma^\rho{}_{\lambda\nu}\frac{\partial x^\lambda}{\partial \tau}\frac{\partial x^\nu}{\partial \tau} = 0 \tag{53}$$

Another way to derive the equation is by finding the minimum value of the distance in curved space.

$$S = \int \sqrt{g_{\mu\nu}\frac{dx^\mu}{d\tau}\frac{dx^\nu}{d\tau}}\,d\tau \tag{54}$$

By varying the above equation and requiring it to be 0, one can compute its minimum value, and after tedious calculation, the geodesic equation can be obtained.

## C Base Graph Neural Network Model

In general, molecule is represented in a graph form. Therefore, in order to handle molecule dataset, it is inevitable to utilize graph neural networks. We chose directional message passing network (DMPNN) (Yang et al., 2019) for our backbone, since it outperforms other GNN architectures in molecular domain. Given a graph, DMPNN initializes the hidden state of each edge $(i, j)$ based on its edge feature $E_{ij}$ with node feature $X_i$. At each step $t$, directional edge summarizes incident edges as a message $m_{ij}^{t+1}$ and updates its hidden state to $h_{ij}^{t+1}$.

$$m_{ij}^{t+1} = \sum_{k \in \mathcal{N}(i)\setminus j} h_{ki}^t \tag{55}$$

$$h_{ij}^{t+1} = \mathrm{ReLU}(h_{ij}^0 + W_e m_{ij}^{t+1}) \tag{56}$$

Where $\mathcal{N}(i)$ denotes the set of neighboring nodes and $W_e$ a learnable weight.he hidden states of nodes are updated by aggregating the hidden states of incident edges into message $m_i^{t+1}$, and passing its concatenation with the node feature $X_i$ into a linear layer followed by ReLU non-linearity

$$m_i^{t+1} = \sum_{j \in \mathcal{N}(i)} h_{ij}^t \tag{57}$$

$$h_i^{t+1} = \mathrm{ReLU}(W_n \mathrm{concat}(X_i, m_i^{t+1})) \tag{58}$$

Similarly, $W_n$ denotes a learnable weight. Assuming DMPNN runs for $T$ timesteps, we use $(X_{out}, E_{out}) = \mathrm{GNN}(A, X, E)$ to denote the output representation matrices containing hidden states of all nodes and edges, respectively (i.e., $X_{out,i} = h_i^T$ and $E_{out,ij} = h_{ij}^T$).

For graph-level prediction, the node representations after the final GNN layer are typically sum-pooled to obtain a single graph representation $h_G = \sum_i h_i$, which is then passed to a FFN prediction layer.

## D Architecture and Hyperparameters

Detailed steps of training *GATE* is described in Algorithm 1. The architecture of our model is composed of five distinct networks and their parameter sizes are depicted in Table 1. As illustrated in Figure 2, one embedding network is shared across tasks, and encoder, transfer, inverse transfer, and head network exists for each task. The embedding network $embedd(\cdot)$ has the DMPNN architecture

---

**Algorithm 1** *GATE*

---

1: Initialize encoder network $f_e$, transfer network $f_t$, inverse transfer network $f_i$, head network $f_h$
   with random parameters $\theta$
2:
3: **for** epoch $i = 1, 2, \ldots n$ **do**
4:     **for** each $t \in Tasks$ **do**
5:       **for** each batch $\mathbf{b} = (x^t, y^t) \in$ dataset $D$ **do**
6:         $a^t \leftarrow embedd(x^t)$
7:         $\{\bar{a}^t\} \leftarrow perturb(a^t)$
8:
9:         $z^t \leftarrow f_e^t(a^t)$
10:        $m^t \leftarrow f_t^t(z^t)$
11:        $\{\bar{z}^t\} \leftarrow f_e^t(\{\bar{a}^t\})$
12:        $\{\bar{m}^t\} \leftarrow f_t^t(\{\bar{z}^t\})$
13:
14:        $L_{reg} \leftarrow MSELoss(y^t, f_h^t(z^t))$
15:        $L_{auto} \leftarrow MSELoss(f_i^t(m^t), z^t)$
16:
17:        **for** each $s \in Subtasks$ **do**
18:          $z^s \leftarrow f_e^s(a^t)$
19:          $m^s \leftarrow f_t^s(z^s)$
20:          $\{\bar{z}^s\} \leftarrow f_e^s(\{\bar{a}^s\})$
21:          $\{\bar{m}^s\} \leftarrow f_t^s(\{\bar{z}^s\})$
22:
23:          $L_{map} \leftarrow L_{map} + MSELoss(y^t, f_h^t \circ f_i^t(m^s))$
24:          $L_{cons} \leftarrow L_{cons} + MSELoss(\{\bar{m}^t\}, m^s)$
25:          $L_{dist} \leftarrow L_{dist} + MSELoss(m^t - \{\bar{m}^t\}, m^s - \{\bar{m}^s\})$
26:        **end for**
27:
28:        Compute $L_{total} = L_{reg} + \alpha L_{auto} + \beta L_{map} + \gamma L_{cons} + \delta L_{dist}$
29:        Update $\theta$ using $L_{total}$
30:       **end for**
31:     **end for**
32: **end for**

---

with depth 2 and converts the input molecule representation $x$ into a new representation $a$ in a common embedding space. We apply perturbation $perturb(\cdot)$ to $a$ for a number of perturbations, which is set to 10 in this paper. All of the perturbed representation $\{\bar{a}\}$ along with $a$ are then fed into the encoder network. The encoder network is composed of backbone network and bottleneck network. Backbone network has the DMPNN architecture with depth 2 and the bottleneck network has an autoencoder structure with MLP layers. The output from the encoder $f_e(a)$ becomes the input to the transfer network and head network. The output of transfer network $f_t(z)$, notated as $m$, is used to calculate consistency loss and distance loss. It is also fed into inverse transfer network, so that the output from inverse transfer network $f_i(m)$ can be used to calculate autoencoder loss. The output from head network $f_h \circ f_i(m)$ is used to calculate regression loss and mapping loss. We trained 600 epochs with batch size 512 while using AdamW (Loshchilov & Hutter, 2017) for optimization with learning rate 5e-5. The hyperparameters for $\alpha, \beta, \gamma, \delta$ are 1, 1, 1, 1 respectively.

Table 1: Network parameters

| network | layer | input, output size | hidden size | dropout |
|---|---|---|---|---|
| backbone | DMPNN | [134,149], 100 | 200 | 0 |
| bottleneck | MLP layer | 100, 50 | 50 | 0 |
| transfer | MLP layer | 50, 50 | 100,100,100 | 0.2 |
| inverse transfer | MLP layer | 50, 50 | 100,100,100 | 0.2 |
| head | MLP layer | 50, 1 | 25,12 | 0.2 |

Table 2: Hyperparameters

| | |
|---|---|
| learning rate | 0.00005 |
| optimizer | AdamW |
| batch size | 512 |
| epoch | 600 |
| # of perturbation | 10 |
| $\alpha, \beta, \gamma, \delta$ | 1, 1, 1, 1 |

# E    DETAILED EXPLANATION OF DATASETS AND EXPERIMENTAL SETUPS

## E.1    DATASETS

Table 3: Detailed information about the datasets.

| name | acronym | source | count | mean | std |
|---|---|---|---|---|---|
| Abraham Descriptor S | AS | Ochem | 1925 | 1.05 | 0.68 |
| Boiling Point | BP | Pubchem | 7139 | 198.99 | 108.88 |
| Collision Cross Section | CCS | Pubchem | 4006 | 205.06 | 57.84 |
| Critical Temperature | CT | Ochem | 242 | 626.04 | 120.96 |
| Dielectric Constant | DK | Ochem | 1007 | 0.80 | 0.41 |
| Density | DS | Pubchem | 3079 | 1.07 | 0.29 |
| Enthalpy of Fusion | EF | Ochem | 2188 | 1.32 | 0.32 |
| Ionization Potential | IP | Pubchem | 272 | 10.00 | 1.63 |
| Kovats Retention Index | KRI | Pubchem | 73507 | 2071.20 | 719.34 |
| Log P | LP | Pubchem | 28268 | 11.17 | 9.89 |
| Polarizability | POL | CCCB | 241 | 0.84 | 0.26 |
| Surface Tension | ST | Pubchem | 379 | 29.01 | 10.36 |
| Viscosity | VS | Pubchem | 294 | 0.47 | 0.87 |
| Heat of Vaporization | HV | Pubchem | 525 | 43.77 | 18.08 |

We used 14 different molecular property datasets from three different open databases, described in Table 3 and below explanations for evaluation of the *GATE*. Before the training process, the data were purified to exclude data with incorrectly specified units, typos, and extreme measurement environments. All datasets were normalized by mean and standard deviation before the training process. We selected 23 pairs of source and target tasks among the 14 datasets, considering the number of data points in each dataset. We also tried to select task pairs with diversity in correlation as shown in the Figure 7 for a fair and unbiased examination. Hereby, we explicitly describe the physical meaning of each dataset.

- **AS** : The solute dipolarity/polarizability.

- **BP** : The temperature at which this compound changes state from liquid to gas at a given atmospheric pressure.

- **CCS** : The effective area for the interaction between an individual ion and the neutral gas through which it is traveling.

- **CT** : The temparature when no gas can become liquid no matter how high the pressure is.

- **DK** : The ratio of the electric permeability of the material to the electric permeability of free space.

- **DS** : The mass of a unit volume of a compound.

- **EF** : The change in enthalpy resulting from the addition or removal of heat from 1 mole of a substance to change its state from a solid to a liquid.

Absolute Corr. Values Between Task Pairs

Figure 7: Pearson correlation between overlapping data points in target dataset and source dataset.

- **IP** : The amount of energy required to remove an electron from an isolated atom or molecule.
- **KRI** : The rate at which a compound is processed through a gas chromatography column.
- **LP** : Logarithmic form of the ratio of concentrations of a compound in a mixture of octanol and water at equilibrium.
- **POL** : The tendency of matter, when subjected to an electric field, to acquire an electric dipole moment in proportion to that applied field.
- **ST** : The property of the surface of a liquid that allows it to resist an external force
- **VS** : A measure of a fluid's resistance to flow.
- **HV** : The quantity of heat that must be absorbed if a certain quantity of liquid is vaporized at a constant temperature.

### E.2 EXPERIMENTAL SETUPS

For evaluation of the *GATE*, we compared the performance of six baseline methods, including STL, MTL, KD, global structure preserving loss based KD (GSP-KD), and transfer learning (retrain all or head network only). All of the baselines share the same base architecture, with a few different details according to methods. The MTL shares parameters of backbone and bottleneck for given two tasks, and only head networks are separated. In the case of the KD, latent vectors from the bottleneck are used as labels for the distillation, and the distillation loss ratio is set to 0.1. Graph Contrastive Representation Distillation (G-CRD) contains contrastive loss as well as the GSP loss (Joshi et al., 2022). However, we only adopt GSP loss since the contrastive loss term is not applicable for regression tasks. For the GSP-KD, node features from the last layer of the backbone network are used to calculate pairwise distances, which are the labels of the distillation process. The loss ratio of the distillation process of the GSP is also set to 0.1. The maximum epoch is set to be 600, and the best models are selected by early stopping.

## F EXPERIMENTAL RESULTS

We express explicit test results in this section. A total of 23 task pairs from 14 distinct datasets were tested thoroughly with seven different models. Four tables are depicted to show the full experimental results. The best result is emphasized by bold and underlined on each individual result, and the second-best result is underlined. The *GATE* outperforms other conventional methods by a noticeable margin. In a random split setup, the *GATE* wins 52.17% out of total tasks, and for up to second, the *GATE* wins 78.26% out of total. In scaffold setup, the *GATE* wins 56.52% and 78.26% respectively.

Table 4: Random Split Result (part 1)

| Tasks | GATE | | STL | | MTL | | KD | |
|---|---|---|---|---|---|---|---|---|
| | RMSE | STD | RMSE | STD | RMSE | STD | RMSE | STD |
| hv ← ds | **0.9221** | 0.0612 | 0.9574 | 0.0519 | 0.9782 | 0.0782 | 1.3726 | 0.2930 |
| as ← bp | 0.4583 | 0.0193 | 0.5125 | 0.0085 | 0.4370 | 0.0119 | 0.5426 | 0.0335 |
| ds ← kri | **0.4145** | 0.0172 | 0.4154 | 0.0045 | 0.4172 | 0.0102 | 0.4403 | 0.0119 |
| hv ← vs | **0.9116** | 0.0522 | 0.9574 | 0.0519 | 0.9700 | 0.1052 | 1.1995 | 0.1419 |
| vs ← hv | **0.5471** | 0.0719 | 0.5947 | 0.0357 | 0.5535 | 0.0353 | 0.5878 | 0.0264 |
| st ← as | **0.6689** | 0.0413 | 0.9902 | 0.0729 | 1.0272 | 0.0244 | 1.1601 | 0.0396 |
| ds ← lp | **0.4046** | 0.0142 | 0.4154 | 0.0045 | 0.4133 | 0.0135 | 0.4378 | 0.0086 |
| pol ← ds | 0.3431 | 0.0475 | 0.3460 | 0.0291 | 0.4367 | 0.1213 | 0.3089 | 0.0270 |
| vs ← bp | **0.4457** | 0.0151 | 0.5947 | 0.0357 | 0.4516 | 0.0366 | 0.6076 | 0.0241 |
| dk ← ef | 0.4331 | 0.0140 | 0.4331 | 0.0358 | 0.4498 | 0.0126 | **0.3852** | 0.0238 |
| as ← ccs | **0.4648** | 0.0139 | 0.5125 | 0.0085 | 0.4677 | 0.0220 | 0.5364 | 0.0211 |
| ct ← bp | 0.1742 | 0.0034 | 0.2549 | 0.1247 | 0.1707 | 0.0132 | 0.1690 | 0.0079 |
| st ← ccs | **0.9546** | 0.0452 | 0.9902 | 0.0729 | 1.0361 | 0.0737 | 1.1731 | 0.0730 |
| ccs ← kri | 0.2476 | 0.0034 | 0.2936 | 0.0110 | 0.2524 | 0.0042 | 0.2622 | 0.0117 |
| hv ← bp | **0.7251** | 0.0581 | 0.9574 | 0.0519 | 0.7550 | 0.0432 | 1.1983 | 0.1815 |
| vs ← ccs | 0.5233 | 0.0323 | 0.5947 | 0.0357 | 0.5792 | 0.0228 | 0.6027 | 0.0127 |
| st ← hv | 0.7647 | 0.0622 | 0.9902 | 0.0729 | **0.7179** | 0.0259 | 1.1270 | 0.0184 |
| hv ← ct | 0.9399 | 0.0896 | 0.9574 | 0.0519 | 1.1118 | 0.1633 | 1.5114 | 0.1845 |
| ip ← bp | 0.5476 | 0.0642 | 0.6695 | 0.0660 | 0.6067 | 0.0345 | 0.5624 | 0.0273 |
| hv ← ef | **0.6131** | 0.0966 | 0.9574 | 0.0519 | 0.8296 | 0.0999 | 1.3659 | 0.2587 |
| hv ← kri | **0.5410** | 0.0732 | 0.9574 | 0.0519 | 0.8631 | 0.0354 | 1.3739 | 0.2487 |
| ct ← kri | 0.1658 | 0.0136 | 0.2549 | 0.1247 | 0.1716 | 0.0090 | **0.1586** | 0.0102 |
| ip ← dk | 0.6510 | 0.0381 | 0.6695 | 0.0660 | 0.7083 | 0.0226 | 0.5508 | 0.0100 |
| mean | **0.5592** | 0.0271 | 0.6642 | 0.0320 | 0.6263 | 0.0414 | 0.7667 | 0.0908 |
| | Count | Ratio | Count | Ratio | Count | Ratio | Count | Ratio |
| 1st | 12 | 52.17% | 0 | 0.00% | 1 | 4.35% | 2 | 8.70% |
| 2nd | 18 | 78.26% | 1 | 4.35% | 7 | 30.43% | 5 | 21.74% |

Table 5: Random Split Result (part 2)

| Tasks | GSP-KD | | Transfer Retrain All | | Transfer Retrain Head | |
|---|---|---|---|---|---|---|
| | RMSE | STD | RMSE | STD | RMSE | STD |
| hv ← ds | 0.9321 | 0.0487 | 1.0428 | 0.1165 | 1.1166 | 0.0024 |
| as ← bp | 0.5315 | 0.0151 | **0.4325** | 0.0104 | 0.7712 | 0.0105 |
| ds ← kri | 0.4147 | 0.0063 | 0.4414 | 0.0154 | 0.8842 | 0.0049 |
| hv ← vs | 0.9154 | 0.0130 | 0.9937 | 0.0821 | 1.0091 | 0.0181 |
| vs ← hv | 0.5619 | 0.0223 | 0.5712 | 0.0232 | 0.7215 | 0.0392 |
| st ← as | 0.9938 | 0.0141 | 1.1296 | 0.1302 | 1.0045 | 0.0220 |
| ds ← lp | 0.4106 | 0.0077 | 0.4280 | 0.0136 | 0.9111 | 0.0022 |
| pol ← ds | **0.2603** | 0.0270 | 0.3741 | 0.0303 | 0.9060 | 0.0141 |
| vs ← bp | 0.5932 | 0.0097 | 0.5445 | 0.0239 | 0.7220 | 0.0645 |
| dk ← ef | 0.4230 | 0.0133 | 0.3936 | 0.0164 | 0.9380 | 0.0026 |
| as ← ccs | 0.5457 | 0.0150 | 0.4741 | 0.0148 | 0.9935 | 0.0033 |
| ct ← bp | 0.2018 | 0.0093 | **0.1563** | 0.0044 | 0.6847 | 0.0186 |
| st ← ccs | 0.9595 | 0.0405 | 1.1334 | 0.0687 | 1.1039 | 0.0046 |
| ccs ← kri | 0.2698 | 0.0095 | **0.2273** | 0.0016 | 0.6166 | 0.0567 |
| hv ← bp | 0.9051 | 0.0571 | 0.8267 | 0.0417 | 0.8829 | 0.0499 |
| vs ← ccs | 0.5269 | 0.0167 | **0.4868** | 0.0119 | 0.8684 | 0.0116 |
| st ← hv | 0.9618 | 0.0086 | 1.0290 | 0.0945 | 1.0102 | 0.0138 |
| hv ← ct | **0.9207** | 0.0112 | 1.2072 | 0.0460 | 1.0302 | 0.0186 |
| ip ← bp | **0.4631** | 0.0037 | 0.9816 | 0.2334 | 0.8732 | 0.0293 |
| hv ← ef | 0.8112 | 0.0463 | 1.0818 | 0.1021 | 0.9616 | 0.0478 |
| hv ← kri | 0.9191 | 0.0676 | 0.9080 | 0.0510 | 1.0715 | 0.0145 |
| ct ← kri | 0.2080 | 0.0057 | 0.1661 | 0.0075 | 0.8349 | 0.0279 |
| ip ← dk | **0.5257** | 0.0192 | 0.6099 | 0.0273 | 1.0336 | 0.0085 |
| mean | 0.6198 | 0.0212 | 0.6800 | 0.0540 | 0.9108 | 0.0181 |
| | Count | Ratio | Count | Ratio | Count | Ratio |
| 1st | 4 | 17.39% | 4 | 17.39% | 0 | 0.00% |
| 2nd | 10 | 43.48% | 5 | 21.74% | 0 | 0.00% |

Table 6: Scaffold Split Result (part 1)

| Tasks | GATE | | STL | | MTL | | KD | |
|---|---|---|---|---|---|---|---|---|
| | RMSE | STD | RMSE | STD | RMSE | STD | RMSE | STD |
| hv ← ds | 0.6939 | 0.0996 | 0.6744 | 0.1079 | 0.6465 | 0.0776 | **0.5920** | 0.0466 |
| as ← bp | **1.0495** | 0.0256 | 1.2828 | 0.0724 | 1.1677 | 0.1068 | 1.3580 | 0.0136 |
| ds ← kri | **0.4395** | 0.0108 | 0.4477 | 0.0052 | 0.4849 | 0.0061 | 0.5409 | 0.0480 |
| hv ← vs | 0.7174 | 0.0796 | 0.6744 | 0.1079 | 0.9954 | 0.2059 | 0.8948 | 0.2294 |
| vs ← hv | **0.6120** | 0.0639 | 0.9816 | 0.1267 | 0.8535 | 0.0558 | 1.2597 | 0.3638 |
| st ← as | **0.7540** | 0.0660 | 0.8041 | 0.1062 | 1.0254 | 0.0251 | 1.7083 | 0.1608 |
| ds ← lp | **0.4049** | 0.0102 | 0.4477 | 0.0052 | 0.4517 | 0.0184 | 0.5221 | 0.0328 |
| pol ← ds | 0.9040 | 0.0852 | 0.9604 | 0.1056 | 1.4198 | 0.0796 | 1.3309 | 0.1998 |
| vs ← bp | 0.6121 | 0.0297 | 0.9816 | 0.1267 | **0.5686** | 0.0276 | 0.9371 | 0.2386 |
| dk ← ef | 0.7122 | 0.0545 | 0.7028 | 0.0391 | 0.6549 | 0.0210 | 0.8189 | 0.0462 |
| as ← ccs | 1.1313 | 0.0496 | 1.2828 | 0.0724 | **1.1197** | 0.0558 | 1.3773 | 0.0781 |
| ct ← bp | **0.3883** | 0.0203 | 1.4436 | 0.1150 | 0.4359 | 0.0126 | 1.2459 | 0.1199 |
| st ← ccs | **0.7281** | 0.0586 | 0.8041 | 0.1062 | 0.9905 | 0.0737 | 1.5402 | 0.1418 |
| ccs ← kri | **0.5292** | 0.0094 | 0.5489 | 0.0107 | 0.5297 | 0.0083 | 0.5534 | 0.0190 |
| hv ← bp | 0.4821 | 0.0132 | 0.6744 | 0.1079 | **0.4668** | 0.0169 | 0.6271 | 0.0868 |
| vs ← ccs | **0.6126** | 0.0671 | 0.9816 | 0.1267 | 0.8186 | 0.0790 | 1.3034 | 0.5354 |
| st ← hv | **0.7209** | 0.0412 | 0.8041 | 0.1062 | 0.7237 | 0.0276 | 1.5256 | 0.1906 |
| hv ← ct | 0.6579 | 0.0678 | 0.6744 | 0.1079 | 0.6633 | 0.0660 | 0.7925 | 0.2694 |
| ip ← bp | 0.4668 | 0.0179 | 0.5780 | 0.1475 | 0.5540 | 0.0587 | **0.4205** | 0.0240 |
| hv ← ef | 0.6406 | 0.0335 | 0.6744 | 0.1079 | 0.7879 | 0.0643 | 0.6773 | 0.1553 |
| hv ← kri | **0.5084** | 0.0264 | 0.6744 | 0.1079 | 0.6204 | 0.0269 | 0.6710 | 0.1524 |
| ct ← kri | **0.3902** | 0.0140 | 1.4436 | 0.1150 | 0.5173 | 0.0927 | 1.3392 | 0.1076 |
| ip ← dk | **0.4335** | 0.0119 | 0.5780 | 0.1475 | 0.5335 | 0.1016 | 0.4975 | 0.0769 |
| mean | **0.6343** | 0.0270 | 0.8313 | 0.0408 | 0.7404 | 0.0441 | 0.9797 | 0.1218 |
| | Count | Ratio | Count | Ratio | Count | Ratio | Count | Ratio |
| 1st | 13 | 56.52% | 0 | 0.00% | 3 | 13.04% | 2 | 8.70% |
| 2nd | 18 | 78.26% | 3 | 13.04% | 9 | 39.13% | 2 | 8.70% |

Table 7: Scaffold Split Result (part 2)

| Tasks | GSP-KD | | Transfer Retrain All | | Transfer Retrain Head | |
|---|---|---|---|---|---|---|
| | RMSE | STD | RMSE | STD | RMSE | STD |
| hv ← ds | 0.7606 | 0.0810 | 0.8659 | 0.0788 | 0.9584 | 0.0339 |
| as ← bp | 1.2340 | 0.0294 | 1.1478 | 0.0264 | 1.0935 | 0.0079 |
| ds ← kri | 0.4467 | 0.0104 | 0.8753 | 0.1134 | 1.0928 | 0.0482 |
| hv ← vs | **0.6536** | 0.0345 | 0.7520 | 0.1666 | 0.7924 | 0.0595 |
| vs ← hv | 0.6377 | 0.0253 | 0.9217 | 0.1575 | 0.9179 | 0.0539 |
| st ← as | 0.9335 | 0.0954 | 1.2604 | 0.0946 | 1.0780 | 0.0613 |
| ds ← lp | 0.4685 | 0.0111 | 0.4664 | 0.0121 | 1.0410 | 0.0026 |
| pol ← ds | **0.8475** | 0.0627 | 1.0385 | 0.2146 | 1.3204 | 0.0491 |
| vs ← bp | 0.6599 | 0.0204 | 1.1532 | 0.1766 | 1.0135 | 0.0820 |
| dk ← ef | **0.6353** | 0.0171 | 0.7417 | 0.0384 | 0.7963 | 0.0071 |
| as ← ccs | 1.1272 | 0.0778 | 1.2925 | 0.0606 | 1.4530 | 0.0143 |
| ct ← bp | 1.1837 | 0.0586 | 0.5644 | 0.053 | 0.9347 | 0.0316 |
| st ← ccs | 0.7344 | 0.0187 | 0.9075 | 0.0431 | 1.2596 | 0.0287 |
| ccs ← kri | 0.5356 | 0.0115 | 0.5640 | 0.0137 | 0.7904 | 0.0159 |
| hv ← bp | 0.7403 | 0.0889 | 0.6093 | 0.0422 | 0.8111 | 0.0251 |
| vs ← ccs | 0.8027 | 0.0159 | 0.7271 | 0.0828 | 1.2282 | 0.0243 |
| st ← hv | 0.7417 | 0.0206 | 1.4243 | 0.0627 | 1.0047 | 0.0813 |
| hv ← ct | **0.6428** | 0.008 | 0.9499 | 0.2579 | 0.8089 | 0.0532 |
| ip ← bp | 0.4579 | 0.0207 | 0.4419 | 0.0371 | 0.9704 | 0.0399 |
| hv ← ef | **0.5862** | 0.0375 | 1.0003 | 0.1719 | 0.9503 | 0.0307 |
| hv ← kri | 0.5509 | 0.0252 | 0.6560 | 0.0408 | 0.9998 | 0.0311 |
| ct ← kri | 1.2358 | 0.0373 | 1.1124 | 0.1265 | 1.2769 | 0.0193 |
| ip ← dk | 0.4376 | 0.0255 | 0.5248 | 0.0471 | 1.0165 | 0.0521 |
| mean | 0.7415 | 0.0363 | 0.8694 | 0.0671 | 1.0265 | 0.0217 |
| | Count | Ratio | Count | Ratio | Count | Ratio |
| 1st | 5 | 21.74% | 0 | 0.00% | 0 | 0.00% |
| 2nd | 11 | 47.83% | 2 | 8.70% | 1 | 4.35% |

