# OpenReview forum: "Geometrically Aligned Transfer Encoder for Inductive Transfer in Regression Tasks"
_ICLR.cc/2024/Conference — ICLR 2024 poster_

### Official Review · Reviewer_NCqT · 2023-10-31

**Soundness:** 2 fair
**Presentation:** 3 good
**Contribution:** 2 fair
**Rating:** 5
**Confidence:** 4

**Summary:**

- The paper introduces a novel transfer algorithm called GATE based on Riemannian differential geometry.
- GATE is designed for regression tasks in inductive transfer learning and outperforms conventional methods in molecular property regressions.
- The authors propose a method to match coordinate patches on a manifold using an autoencoder scheme and consistency loss.
- The approach allows for the flow of information from the source domain to the target domain, improving the model's generalization capabilities.

----
I would like to thank the authors for the detailed rebuttal.
After reading the rebuttal and other reviewers point, I would like to increase the original score.

**Strengths:**

- Experiment demonstrates superior performance of GATE in extrapolation tasks, with 14.3% lower error in scaffold split compared to conventional methods.
- This paper shows stable underlying geometry of GATE's latent space and robust behavior in the presence of data corruption.
- This paper provides ablation studies and further analysis to understand the role of distance loss and the stability of the latent space.

**Weaknesses:**

- The experiments are based on 14 molecular datasets and most counts are limited (largest one contains 73k count, while smallest only has 241). This limits the generalization of such method on other large dataset.
- Lack of analysis on the computational efficiency or scalability of the GATE algorithm.
- The proposed method, in abstract, is an encoder-decoder-based method, which is kind of trivial in the transfer learning setup.

**Questions:**

- The distance loss is simplified as equation 12, which is basically Euclidean distance. This is contradict to the manifold setup. Can the author discussed more about the distance loss, which seems to be the major difference the author proposed compared with other existing methods.
- The ablation study shows that the problem is an overfitting problem. So it basically means the data is not enough/ early stopping should help. So the real contribution of such method is slightly not distinguishable.
- Some tiny piece:
  - 80-20 split of train and test, is not 4-fold cross-validation.
  - Figure 3 shows GATE as circle, does this mean all equal or it takes GATE as reference?

---

> ### Author Response · Authors · 2023-11-15
>
> Weaknesses:
> The experiments are based on 14 molecular datasets and most counts are limited (largest one contains 73k count, while smallest only has 241). This limits the generalization of such method on other large dataset.
>
> - The reviewer appears to have a misunderstanding about transfer learning in general contexts. Transfer learning is a methodology aimed at transferring useful information from datasets with abundant or robust model knowledge to tasks where data is scarce or learning is challenging. To test this, it is natural to use datasets with a limited amount of data. Furthermore, the applicability of a model is not restricted by what dataset we used to demonstrate the performance. Given that the primary contribution of this paper lies in enhancing generalization ability, the reviewer's comment appears to be unreasonable.
>
>
> Lack of analysis on the computational efficiency or scalability of the GATE algorithm.
>
> - For the same reasons mentioned above, I do not consider scalability to be a central issue in this paper. However, upon closer examination of this aspect, fundamentally, the scalability is similar to training two single-task models. The difference lies in the additional parameters introduced by the transfer module and inv transfer module. It is not a setup that poses a significant scalability challenge to the extent that it requires a thorough analysis of computational efficiency.
>
>
> The proposed method, in abstract, is an encoder-decoder-based method, which is kind of trivial in the transfer learning setup.
>
> - The reviewer's description is entirely different from the technical aspects of this paper. The use of an encoder-decoder is merely a small part adopted for the implementation of the proposed technique. For more detailed information, please refer to the distance loss section in the general comments.
>
>
> Questions:
> The distance loss is simplified as equation 12, which is basically Euclidean distance. This is contradict to the manifold setup. Can the author discussed more about the distance loss, which seems to be the major difference the author proposed compared with other existing methods.
>
> - It is not contradictory. For more detailed information, please refer to the distance loss section in the general comments.
>
>
> The ablation study shows that the problem is an overfitting problem. So it basically means the data is not enough/ early stopping should help.
> So the real contribution of such method is slightly not distinguishable.
>
> - First and foremost, as is in most standard setups, we applied early stopping method based on the validation loss. This is explicitly stated in Appendix E of the paper. We observed the trend of the validation loss, as shown in Fig 4 of the paper, and there is absolutely no reason not to apply early stopping when we are aware of it.
>
> Furthermore, discussing issues that cannot be addressed by early stopping is a fundamental setup in researching generalization problems. As mentioned in the general comments section regarding molecular property prediction, generalization issues are crucial in deep learning, particularly when obtaining additional data comes with significant costs.
>
> Some tiny piece:
> 80-20 split of train and test, is not 4-fold cross-validation.
>
> - We performed cross-validation by dividing 80% of the training set into 4 folds, using 20% as the validation set for each fold. The statistics for the four models trained in this manner are test for the 20% of distinct test set and recorded in the paper's table.
>
>
> Figure 3 shows GATE as circle, does this mean all equal or it takes GATE as reference?
>
> - It appears that the reviewer may have overlooked Section 4.2 which is a crucial part of the paper. Similarly, it seems that the caption for Fig 3 was also overlooked. The detailed explanation of how the figure was generated is provided in the main text; therefore, additional clarification will not be provided in this comment.

---

### Official Review · Reviewer_nPy4 · 2023-10-31

**Soundness:** 3 good
**Presentation:** 3 good
**Contribution:** 3 good
**Rating:** 6
**Confidence:** 3

**Summary:**

This paper tackles the problem of inductive transfer learning in regression tasks. Assuming that the latent vectors of the model lie on a smooth Riemannian manifold, the paper suggests that for effective transfer learning, source and target tasks need to be mapped to regions with large overlap. The paper describes how diffeomorphisms between pairs of tasks can be learned using parametric encoders and decoders, such that individual data points are confined to a locally flat frame in the overlap region. This is a local Euclidean approximation that helps simplify distance calculation when only small perturbations are concerned.

The proposed method (GATE) adds 3 losses to the regression loss: an auto-encoder loss for the choice of diffeomorphism modeling, a consistency loss that pivots points on the overlapping region, a mapping loss that enforces predictions to be preserved through the transforms, and a distance loss that forces distances between pivot points and perturbations to be equal across tasks. The distance loss can be viewed as a regularizer.

The proposed method is evaluated on a transfer learning task for molecular property prediction and is shown to outperform alternatives in a majority of the transfer tasks considered. The overall RMSE is also significantly lower than the baseline methods.

**Strengths:**

Originality and significance: The Riemannian view of the latent space is likely not new to this work, but two of the loss functions are novel to this work (to the best of my knowledge). The empirical improvement over the baselines in the studied task of molecular property prediction is significant.

Quality and clarity: The paper is overall well motivated. The descriptions are accompanied by formulas and helpful schematic diagrams. The empirical analysis covers several aspects of the proposed solution, including ablation studies to gauge the effect of each part of the loss function.

**Weaknesses:**

Even though the method is likely applicable and useful in other domains, the paper only studies it on the molecular property prediction task. This significantly cuts into the impact of the paper as the results cannot (in good faith) be extrapolated to a completely different domain of tasks.

With a single model architecture for molecular property prediction as the only domain, a more detailed description is missing from the main body of the paper (e.g. input/output/latent dimensions, range of values, SMILES format, DMPNN layers, etc.). The paper is not self-contained when it comes to model components.

The paper is hard to follow in parts, as the overall picture takes more than 4 pages to be completely laid out. I think an overview of all the losses can be included earlier in the paper to help with the flow and clarity of the paper.

Since the heavy machinery of Riemannian geometry is not really used in the paper (apart from freedom for choice of local coordinates and the overall smoothness assumptions), IMO the elaborate dive into the nuances of dealing with geodesics in an arbitrary metric space is unnecessary, if not needlessly confusing. There is no point introducing the Christoffel symbol just to scare off a reader that might think a nice closed-form solution is possible in the generic case. I’d suggest removing derivations and definitions that do not contribute to the flow of the paper.

**Questions:**

Notes and questions:
- The typo in equation 7 was particularly hard to resolve while reviewing this paper. The choice of notation might have contributed to this.
- It’s not clear what you mean by "stable characteristics" in section 5.2. Please elaborate on what makes MTL relatively unstable in this case.
- Lower bounds are not shown in Figure 6.

---

> ### Author Response · Authors · 2023-11-15
>
> Weaknesses:
> Even though the method ~
>
> - As mentioned in the general comments regarding molecular property prediction, such assertions are entirely unfounded. Upon reviewing ICLR 23 papers, it is evident that there are numerous papers solely focused on molecular property prediction, many of which are influential and well-regarded.
>
>
> With a single model architecture for ~
>
> - The detailed information is all provided in Appendix E, and it appears that the reviewer might have overlooked this section. Please refer to that part for comprehensive details.
>
> The paper is hard to follow in parts, ~
>
> - We will gladly update Fig. 2 to include an overview of all the losses. However, due to the short rebuttal period, we plan to address this in future revisions
>
> Since the heavy machinery of Riemannian geometry is ~
>
> - We strongly disagree with the reviewer's comment. As explained in the general comments regarding the distance loss, a fundamental understanding of Riemannian geometry is essential to comprehend the impact of this loss. While some readers may already possess this knowledge, many may not be familiar with it. Therefore, we have opted not to exclude this information from the main text. This section is intended to provide assistance to a broad audience, including the reviewer, in understanding our algorithm. Despite the seemingly straightforward implementation of distance loss, the underlying mathematical background and Riemannian geometry are by no means trivial. Implementing a model is one thing, but understanding why a model works is crucial. To grasp diffeomorphism accurately, one must understand coordinate transformation rules, be aware of covariance in curved space (requiring an understanding of Christoffel symbols), and even discuss Riemannian curvature for a thorough understanding of diffeomorphism invariance in Riemannian geometry. Due to the mathematical complexity, we did not delve into the details of Riemannian curvature in the main text. Therefore, the content is not intended to intimidate the reader; on the contrary, it is an omission of even more complex mathematical details underlying our algorithm. The mention of geodesics in the Conclusion section is included in the Appendix to aid in the understanding of future research directions. All these details are included to ensure the self-containedness of the paper.
>
>
> Questions:
> Notes and questions:
>
> The typo in equation 7 was particularly hard to ~
>
> - We have identified some missing content in certain mathematical expressions, and we are aware that there may be confusion due to the different labels between Fig 2 and Section 3.2 in the main text. To address this, we have uploaded a revised version unifying the formulas and notations in Fig 2 and Section 3.2. Additionally, we have made the overall mathematical expressions and explanations in Section 3.2 more accessible.
>
>
> It’s not clear what you mean by "stable~
>
> - Multi-Task Learning (MTL) and transfer learning can experience negative transfer, leading to potential performance degradation, depending on the types of source and target tasks. This phenomenon is well-documented, as indicated in the paper titled 'A Survey on Negative Transfer,' published in IEEE TRANSACTIONS ON NEURAL NETWORKS AND LEARNING SYSTEMS in 2021. The survey discusses the following points:
> "Multi-task learning solves multiple learning tasks jointly, by exploiting commonalities and differences across them. Similar to TL, it needs to facilitate positive transfer among tasks to improve the overall learning performance on all tasks. Previous studies have observed that conflicting gradients among different tasks may induce NT (also known as negative interference). Various techniques have been explored to remedy negative interference, such as altering the gradients directly, weighting tasks, learning task relatedness, routing networks, and searching for Pareto solutions, etc."
>
> Furthermore, the relationship between the alignment of the latent space and performance based on target values can be found in the paper titled 'Automatic Chemical Design Using a Data-Driven Continuous Representation of Molecules,' published in ACS Cent. Sci. 2018. In our paper, additional ablation studies were conducted, confirming that there is deformation in the latent space when negative transfer occurs. As shown in the Fig. 5, it becomes evident that the shape of the latent space in GATE remains consistently maintained according to the source task. In contrast, MTL exhibits significant deformations in the shape depending on the source task.
>
>
> Lower bounds are not shown in Figure 6.
>
> - We have opted for a notation that represents the overall error in the upper part, rather than separately calculating and indicating upper and lower errors. Therefore, the lower error range is identical to the upper error range, and in the current experimental setup, we believe there is no need to distinguish between upper and lower errors.

---

> > ### Comment · Reviewer_nPy4 · 2023-11-20
> > **Revisions look good**
> >
> > The revised submission fixes several major issues discussed in the reviewers and is much easier to follow with the cleaned up notation. I'm updating the overall score accordingly.
> >
> > - Re use case: I still think that any claims beyond the application of method to molecular property prediction is unsupported by the evidence put forth by the paper. This particular use case is important and relevant to the community, but I would like the paper to avoid any hard claims that the method is applicable elsewhere. Any such claim should be clearly stated as an educated guess since the analysis is not covered by the paper.
> >
> > - Riemannian geometry discussions: Other reviewers have mirrored some of my initial discussion that the overly compressed overview of the unused mathematical machinery is not helpful with the flow of the paper and is unlikely to help readers that are not already familiar with the topic. I suggest moving unused math to the appendix so that the discussion is not compressed to the point that is not easy to follow. IMO the space in the main body of the paper can be better used to for discussions and additions suggested by the reviews, e.g. more details about the experimental setup or an expanded overview of the molecular property prediction task.

---

### Official Review · Reviewer_Gg1C · 2023-11-07

**Soundness:** 2 fair
**Presentation:** 2 fair
**Contribution:** 3 good
**Rating:** 8
**Confidence:** 2

**Summary:**

This paper proposes an approach to improve multi-task learning by training separate learners for each task but transferring latent representations between them under the assumption that the latent spaces can be modeled as Riemannian manifolds.  This is accomplished by introducing encoders and decoders between the latent spaces of the source and target task, and training by use of the original loss function plus a cycle consistency loss for the encoder-decoder pair for each direction, and a distance loss based on perturbations to the input to ensure that the latent space mapping from source to target preserves the local metric.  The resulting method, GATE, is applied to a set of molecular prediction tasks, and compared to existing methods such as single-task learning, multi-task learning, transfer learning, knowledge distillation, and global structure-preserving knowledge distillation.

**Strengths:**

This paper proposes an interesting approach to transfer learning, and the contribution of each proposed feature (consistency loss, distance loss) is analyzed in an ablation study.  It is very interesting that the distance loss can prevent overfitting.  The choice of a dataset with 14 different tasks is suitable for a multi-task setup.  The use of two different random splits shows careful consideration of the testing setup.  The graphical check of the latent space across tasks helps builds confidence in the method.  The results are reported in detail in the appendix which allows deeper analysis by interested readers.

**Weaknesses:**

The idea of enforcing cycle-consistency is not new, and it seems appropriate to cite related literature on cycle-consistency within transfer learning such as CyCADA (Hoffman et al. 2018).

I found it difficult to follow the notation introduced to explain the method given the lack of explanation on the notations.  More details are needed to describe the method precisely.

It is not clear to me how distance loss helps prevent overfitting.  A toy example would help with the intuition.

The fact that only chemistry applications were considered limits the generality of this paper to folks working in other domains.

It is not clear how much the concepts of Riemannianian geometry actually add to the paper (what surprising results or insights depend on deep findings from differential geometry?) but instead seem to obfuscate the relatively simple and intuitive ideas which are implemented in the method GATE.

**Questions:**

1. What surprising results or insights depend on deep findings from differential geometry?
2. To what degree does GATE succeed at learning a cycle-consistent and metric-preserving map?
3. What are limitations of the method?  Under what conditions would it do worse than conventional multi-task learning?
4. What is X'' in equation (7)?
5. What are the details for how STL, MTL, transfer learning, KD and GSP-KD were trained?
6. Could the fact that distance loss helps prevent overfitting be related to training with adversarial examples?
7. What happens if you only remove the cycle-consistency loss, but not the distance loss?

---

> ### Author Response · Authors · 2023-11-15
>
> Weaknesses:
> The idea of enforcing cycle-consistency ~
>
> - We acknowledge the pre-existence of the concept of cycle-consistency, and this aspect is entirely unrelated to the contributions of this paper. The contribution of this paper lies in the application of distance loss to enhance generalization, as detailed explanations can be found in the general comments section
>
>
> I found it difficult to follow the notation introduced~
>
> - We have identified some missing content in certain mathematical expressions, and we are aware that there may be confusion due to the different labels between Fig 2 and Section 3.2 in the main text. To address this, we have uploaded a revised version unifying the formulas and notations in Fig 2 and Section 3.2. Additionally, we have made the overall mathematical expressions and explanations in Section 3.2 more accessible.
>
>
> It is not clear to me how distance loss ~
>
> - Please refer to the explanation in the general comments section regarding the distance loss.
>
>
> The fact that only chemistry applications ~
>
> - Please refer to the explanation in the general comments section regarding the molecular property prediction.
>
>
> It is not clear how much the concepts of Riemannianian ~
>
> - Please refer to the explanation in the general comments section regarding the distance loss.
>
>
> Questions:
> What surprising results or insights depend ~
>
> - Please refer to the explanation in the general comments section regarding the distance loss.
>
>
> To what degree does GATE succeed at ~
>
> - Upon investigation, it has been observed that the average standard deviation of the encoded vectors is around 0.07, while consistency is learned at an error level of 0.01. Additionally, the standard deviation of the reference distance, taken as a benchmark, is approximately 3.1*E-06, whereas the learned distance is at an error level of 7.6 * E-08. This confirms that training process is valid enough, and these values may vary depending on the scaling factor of the loss parameter.
>
>
> What are limitations of the method? Under ~
>
> - It seems challenging to pinpoint a specific scenario where the performance is notably inferior to MTL. However, it should be noted that, in comparison to MTL, the model complexity increases, and there is an increase in the number of parameters, leading to a slightly slower learning speed (in linear order).
>
>
> What is X'' in equation (7)?
>
> - We have identified some missing content in certain mathematical expressions, and we are aware that there may be confusion due to the different labels between Fig 2 and Section 3.2 in the main text. To address this, we have uploaded a revised version unifying the formulas and notations in Fig 2 and Section 3.2. Additionally, we have made the overall mathematical expressions and explanations in Section 3.2 more accessible.
>
>
>
> What are the details for how STL, MTL, ~
>
> - In Appendix E, detailed information about the network parameters, hyperparameters, and architecture are explicitly provided. As described in Appendix E.2, for a fair experiment, the base architecture was consistently employed across all methods. Additionally, differences in training details among these methods have been carefully documented. We feel that all necessary elements, such as epochs, learning rate, etc., are already included in the paper. If there are specific additions deemed necessary, please specify, and we will refer to it for revision
>
>
> Could the fact that distance loss helps prevent ~
>
> - The mapping loss is computed only for input points, and there is no downstream task for neighboring points. Therefore, adversarial examples do not exist in the training setup. For detailed information on the distance loss, please refer to the general comments. As we are confused about what adversarial expamples are in this context, it may be worth mentioning that, similar to the response to another reviewer's question, distance loss has no relevance to contrastive learning.
>
>
> What happens if you only remove the ~
>
> - We conducted experiments by removing only the consistency loss. In this case, information about the input point is absent in the distance loss, it is impossible to pin point the reference point to align the latent spaces. Therefore, theoretically, aligning the source and target latent spaces is not possible, and as a result, we expected a performance degradation. Indeed, during the actual experiment, as anticipated, overfitting occurred right after 50 epochs of training, leading to a decline in performance from around 0.2 MSE to over 0.4, which is a similar scale in the mapping only case. Since this case is theoretically trivial enough, we choose not to include it in the manuscript. However, since the comment system in openreview seems to not accept any images, we added the experiment graph to the supplementary materials.

---

> ### Comment · Reviewer_Gg1C · 2023-11-15
> **Revision much clearer**
>
> Thanks for straightening out the notation in the paper.  Actually being able to understand the method deepens my appreciation of it.  I am happy to increase my presentation rating from Poor to Fair and raise my score from 5 to 8.
>
> As both nPy4 and myself think that the inclusion of detailed discussions of Riemannian geometry are helpful to this paper, I will respond to your comment where you state "As explained in the general comments regarding the distance loss, a fundamental understanding of Riemannian geometry is essential to comprehend the impact of this loss."  It may be true that exposure to Riemannian geometry gives deep insight to the method, but a practitioner does not need formal mathematical training to appreciate that the distance loss helps constrain the mapping in a way that prevents overfitting.  Also, it is worth considering the practicality of attempting to give a crash course on differential geometry within a few pages.  Assuming the possibility of a reader who lacks exposure to differential geometry but is interested in learning more about the concepts that led to the paper, such would be better served by looking up a good tutorial rather than trying to decipher the highly compressed overview given in the paper.  Hence, it would be far better to direct readers to such material rather than to overload the main paper with background material which is unlikely to be of much use to either the beginner (who cannot understand it) nor the expert (for whom it is redundant.)

---

### Official Review · Reviewer_6Cr1 · 2023-11-09

**Soundness:** 3 good
**Presentation:** 2 fair
**Contribution:** 3 good
**Rating:** 6
**Confidence:** 3

**Summary:**

The authors describe a novel formulation of regression transfer learning by embedding the latent space in a Riemannian manifold, allowing the notion of consistency across tasks and the mapping of different points together.

**Strengths:**

- Novel regularization procedure which also has the potential of being used outside of the scope of this paper.
- Superior performance when compared to other methods for transfer learning.
- Intuitive idea and easy to implement.
- Good experimental section, with a nice exploration of overfitting.

**Weaknesses:**

- The writing quality needs to be improved, there are both distracting grammar issues and, more importantly, the mathematical formulation of the method and description of the prerequisites for understanding this work have not been adequately presented.
- Section 5.2 is not well-supported, specifically the assertion "Ideally, if a model is well-guided by the right information and regularized properly, the overall geometry of the latent space may remain stable and not depend on the type of source tasks. However, if the target task is overwhelmed by the source task and regularization is not enough, latent space will be heavily deformed according to the source tasks" which needs more detail, or a few corroborating citations.

**Questions:**

- In Section 5.3, it seems that using the word "significant" when comparing the GATE and MTL results is not precise enough. Can the authors add a statistical test?
- In the same section, I would also be interested in re-running this experiment with higher deviations, both of the same sign as the original data, just made more extreme (a data point that is 2 sigma away from the mean is made to be 10 sigma away) and the opposite sign (2 sigma more than the mean is changed to 10 sigma less than the mean). Is there a point at which MTL and GATE have more similar performance, or does the gap increase?
- How is this approach connected to contrastive learning and metric learning?

---

> ### Author Response · Authors · 2023-11-15
>
> Weaknesses:
> The writing quality needs to be improved,~
>
> - We have identified some missing content in certain mathematical expressions, and we are aware that there may be confusion due to the different labels between Fig 2 and Section 3.2 in the main text. To address this, we have uploaded a revised version unifying the formulas and notations in Fig 2 and Section 3.2. Additionally, we have made the overall mathematical expressions and explanations in Section 3.2 more accessible.
>
>
> Section 5.2 is not well-supported, specifically the assertion ~
>
> - It is known that MTL or transfer learning may experience negative transfer depending on the types of source and target tasks, leading to performance degradation. The paper titled "A Survey on Negative Transfer," published in IEEE TRANSACTIONS ON NEURAL NETWORKS AND LEARNING SYSTEMS in 2021, provides insights into this phenomenon. According to the survey,
>
> "Multi-task learning solves multiple learning tasks jointly, by exploiting commonalities and differences across them. Similar to TL, it needs to facilitate positive transfer among tasks to improve the overall learning performance on all tasks. Previous studies have observed that conflicting gradients among different tasks may induce NT (also known as negative interference). Various techniques have been explored to remedy negative interference, such as altering the gradients directly, weighting tasks, learning task relatedness, routing networks, and searching for Pareto solutions, etc."
>
> conflicting gradients among different tasks may induce negative transfer, and various techniques have been explored to address this issue.
>
> Additionally, the relationship between the alignment of the latent space based on target values and performance can be found in "Automatic Chemical Design Using a Data-Driven Continuous Representation of Molecules," published in ACS Cent. Sci. 2018. We have added references to these findings in the revised manuscript.
>
>
> Questions:
> In Section 5.3, it seems that using the word ~
>
> - We performed a significance analysis based on p-values for the entire set of 23 tasks in the experiment, comparing the performance difference between the GSP-KD method (2nd best performance) and GATE. In random split, GATE showed significant improvement in 11 tasks with one star or more, while GSP-KD outperformed in 3 tasks. For three stars or more, GATE excelled in 11 tasks, and GSP-KD excelled in 1 task. In scaffold split, GATE showed significant improvement in 9 tasks with one star or more, while GSP-KD outperformed in 2 tasks. For three stars or more, GATE excelled in 7 tasks, and GSP-KD showed no instances of significant improvement. Overall, as there are numerous tasks with three stars or more, indicating a substantial difference, we have chosen not to make specific modifications regarding significance in the manuscript.
>
>
> In the same section, I would also be interested in ~
>
> - Through experimentation, we increased the corruption level to 10 sigma, and for heat of vaporization, GATE exhibited a 11.5% better performance than MTL, which is similar to the 11.7% improvement shown in Fig 6. For collision cross section, at 10 sigma, GATE showed approximately 6.3% better performance than MTL, slightly decreasing from the 10.3% at 2 sigma. While we could not thoroughly investigate due to time constraints, even in extreme corruption scenarios, GATE demonstrated better performance compared to MTL.
>
>
> How is this approach connected to contrastive learning and metric learning?
>
> - There is no connection between the proposed GATE algorithm and contrastive learning or metric learning. Contrastive learning deals with the distance between latent vectors from two different models, whereas the proposed method aims to align the shape of the latent space by comparing the distances between neighboring points and input points for each source and target model. Therefore, there is no common concept between these methodologies. Further details have been provided in the general comments section.

---

> > ### Comment · Reviewer_6Cr1 · 2023-11-18
> > **Thanks for your response**
> >
> > The paper is now clearer and includes statistical analysis for the experiments done. I have thus raised my score from 5 to 6 and think that it should be accepted.

---

### Author Response · Authors · 2023-11-15
**Commen Comment on the Distance Loss**

Upon careful examination of the review comments, it appears that all reviewers are having difficulty of understanding the core ideas of this algorithm and underlying mathematics. As a response, common comments are provided for clarification.

This algorithm starts with the assumption that the latent space of each task is composed of a Riemannian Curved Manifold. As explained in Appendix B, Riemannian manifolds have the advantage of diffeomorphism invariance, providing flexibility in coordinate choice. Furthermore, this property allows arbitrary points on the Riemannian manifold to be locally transformed into a flat space, making it a crucial aspect of the algorithm.

The goal of this algorithm is to effectively integrate information from two different tasks, ensuring general predictive performance even with limited data in each task. This is achieved through the following steps:

(1) Train models for each task through conventional regression learning.

(2) Simultaneously, train an autoencoder to have the same coordinate frame for the latent of each task.

(3) Enforce the encoded latent vectors from the autoencoder to be in a locally flat frame through distance loss.

(4) Finally, transfer appropriately derived latent from different tasks to transform into the coordinate of the target task and conduct regression through mapping loss.


The distance loss, particularly (3), is a crucial aspect, and various ablation studies supporting this are conducted, as outlined in Section 5. Additionally, the mathematical derivations of the Riemannian manifold basics in the appendix are essential to understand the underlying mechanism of this distance loss.

The distance loss, as stated in the main text and the pseudocode in Appendix lines 6-12, 18-21, 25, can be described as follows:

(1) Obtain latent vectors by inputting the given input data point and its infinitesimal neighbors into the source and target models.

(2) Obtain latent vectors corresponding to input data points and neighbors for both source and target models.

(3) Send these obtained latent vectors through respective transfer modules to a common flat coordinate. (Total 4 types of latent vectors transferred: Source input latent, source neighbor latent, target input latent, target neighbor latent)

(4) Calculate the distance between "the input point and its neighbors for the source", and calculate the distance for the target by same process. (Obtain Source distance and target distance)

(5) Apply loss to make the source distance and target distance equal.


The consistency loss, ensuring that each source and target input point becomes the same, does not guarantee sameness in the region without given input points in the latent space. Therefore, distance loss is necessary, providing a powerful constraint by making not only input points but also the surrounding coordinate areas aligned together. Fig. 4 demonstrates the train-validation curve of the model with the addition of consistency and distance loss on top of the mapping loss, clearly resolves overfitting issues when distance loss is added and even surpasses the performance of the MTL model.

In general, distance in curved space is defined by geodesics, requiring solving the geodesic equation, as discussed in Appendix B.4. However, choosing diffeomorphism to a locally flat space allows simplification of the distance between input points and their infinitesimal local neighbors to Euclidean. Sections 3.1 and 3.2 in the main text elaborate on this explanation. These sections reveal that using Euclidean for distance loss calculation yields the following effects:

(1) Latent obtained through the transfer module is in a locally flat space.

(2) The metric can satisfy the triangular inequality that it must meet.

(3) The possitive definite property of metric is manifest infinitesimally.


Therefore, when the latent space is in a Riemannian manifold, it functions well, and the transferred latent space can also be forced into a locally flat space.
Ultimately, the distance loss enforces the latent spaces of the source and target models to be similar in shape, leading to a generalization effect. Moreover, despite having complex geometrical concepts, it can be implemented with a simple combination of well-known deep learning models and straightforward loss terms which is the art of this algorithm.

---

### Author Response · Authors · 2023-11-15
**Common Comment on the Molecular Property Prediction**

The application of proposed technique in this paper is not limited to molecular data. Nonetheless, the prediction of molecular properties constitutes a broad and crucial task within the machine learning society. An exemplary instance is the paper titled "Pre-training via Denoising for Molecular Property Prediction," which was notably accepted in the top 25% at ICLR23. Numerous other papers have been presented, addressing various techniques for solving the significant task of molecule property prediction.

The importance of AI in molecular property prediction is recognized not only in AI conferences but also across the broader research area. This acknowledgment is evident in a recent paper published in Nature titled "Scientific discovery in the age of artificial intelligence."

Furthermore, recent research related to molecular data has found utility in various fields. For example, the DMPNN (refer to Appendix C), a message-passing method within graph neural networks, is not only a valuable algorithm designed for molecular property prediction but is also applied across diverse graph neural network applications beyond the molecular domain. Since our model is input data-agnostic, theoretically, there are no constraints on applying it to data from other fields if appropriate regression tasks exist.

In addition, one of the critical bottlenecks in Molecule Property Prediction is generalization. As highlighted in the paper "Molecular Property Prediction: A Multilevel Quantum Interactions Modeling Perspective" published in AAAI 2019:
"In the field of chemistry, the set of all possible molecules in unexplored regions is called chemical space. One of the famous chemical space projects collected 166.4 billion molecules while merely 134k samples of them were labeled. Therefore, the generalization ability of enabling accurate prediction with the limited dataset is indeed essential in our task."

The importance of generalization in this field is further emphasized by Yoshua Bengio's paper presented at the ICLR22 MLDD workshop, titled "Evaluating Generalization in GFlowNets for Molecule Design."

In this paper, we propose a novel algorithm, including distance loss, to enhance generalization and provide experimental validations across various scenarios.

Finally, although we sought to test our approach on datasets from other fields, we encountered challenges in finding datasets suitable for "inductive regression transfer", as most available datasets are focused on classification tasks. We welcome any insights or information on relevant datasets for further exploration.

---

### Meta-Review · Area_Chair_mgiT · 2023-12-05

**Metareview:**

This paper proposes a new transfer learning approach based on Riemannian differential geometry with Geometrically Aligned Transfer Encoders. An experimental evaluation on graph datasets from the chemical domain is provided.

On the positive side the paper is interesting, it provides new transfer learning method, the application to the chemistry domain is interesting and the paper is well motivated.
 On the negative side, the paper lacks clarity in some parts, some elements in the contribution are a bit incremental, and the experimental evaluation is limited to the chemical domain which limits its scope, some parts are unclear/not sufficiently explained and need an improved rewriting.

During rebuttal, authors have provided multiple answers to the reviewers. Along with specific answers, two general answer and a new revision have been posted. The revision has notably helped to solve some issues related to mathematical formulations. After the rebuttal 3 reviewers agreed that the work done during rebuttal was of good quality and they increase their score to give at least a 6.

Overall, the contribution appears interesting and solid.
I propose acceptance.

**Justification For Why Not Higher Score:**

The global evaluation is positive but not strongly positive.

**Justification For Why Not Lower Score:**

3 reviewers have increased their score to give at least a 6.

---

### Decision · Program_Chairs · 2024-01-16

Accept (poster)